# Beyond the Seen: Bounded Distribution Estimation for Open-Vocabulary Learning

**Xiaomeng Fan**[1*] **Yuchuan Mao**[1*] **Zhi Gao**[1†]
**Yuwei Wu**[1,2]    **Jin Chen**[1]    **Yunde Jia**[2,1†]
[1]Beijing Key Laboratory of Intelligent Information Technology,
School of Computer Science & Technology, Beijing Institute of Technology
[2]Guangdong Laboratory of Machine Perception and Intelligent Computing,
Shenzhen MSU-BIT University
`https://github.com/Beyond-the-Seen-NeurIPS`

## Abstract

Open-vocabulary learning requires modeling the data distribution in open environments, which consists of both seen-class and unseen-class data. Existing methods estimate the distribution in open environments using seen-class data, where the absence of unseen classes makes the estimation error inherently unidentifiable. Intuitively, learning beyond the seen classes is crucial for distribution estimation to bound the estimation error. We theoretically demonstrate that the distribution can be effectively estimated by generating unseen-class data, through which the estimation error is upper-bounded. Building on this theoretical insight, we propose a novel open-vocabulary learning method, which generates unseen-class data for estimating the distribution in open environments. The method consists of a class-domain-wise data generation pipeline and a distribution alignment algorithm. The data generation pipeline generates unseen-class data under the guidance of a hierarchical semantic tree and domain information inferred from the seen-class data, facilitating accurate distribution estimation. With the generated data, the distribution alignment algorithm estimates and maximizes the posterior probability to enhance generalization in open-vocabulary learning. Extensive experiments on 11 datasets demonstrate that our method outperforms baseline approaches by up to 14%, highlighting its effectiveness and superiority.

## 1 Introduction

Open-vocabulary learning, an increasingly prominent task in computer vision, aims to recognize objects for both seen and unseen classes in open environments [65, 49, 80]. Effectively modeling the data distribution in open environments requires capturing both seen-class and unseen-class distributions. However, existing methods estimate open-environment distributions only based on seen-class data [48, 78, 77, 24, 25], and the absence of unseen classes makes it challenging to obtain accurate distribution estimation in open environments.

In this paper, we study how to learn beyond the seen classes in open environments. We derive distribution estimation theorems that prove the distribution can be estimated by generating unseen-class data, with an upper bound on the estimation error. Furthermore, these theorems reveal that narrowing the distribution gap between seen-class and generated unseen-class data tightens this upper bound, leading to more precise estimation. Motivated by these theoretical insights, it is desirable to generate unseen-class data that closely aligns with the seen-class data distribution, enabling a

---

[*]Equal contribution.
[†]Corresponding authors.

39th Conference on Neural Information Processing Systems (NeurIPS 2025).

more accurate distribution estimation in open environments. To this end, we propose a novel open-vocabulary learning method, which generates unseen-class data for distribution estimation in open environments. The method is composed of a class-domain-wise data generation pipeline and a distribution alignment algorithm.

The class-domain-wise data generation pipeline consists of three key components: a hierarchy-guided unseen class predictor, a caption-based domain information generator, and a text-to-image model. These components collaboratively generate unseen-class data while minimizing distribution differences from the seen-class data. Specifically, the hierarchy-guided unseen class predictor unseen class predictor leverages a hierarchical semantic tree, constructed from seen classes (leaf nodes) and their superclasses (parent nodes). This tree is expanded with candidate unseen classes sourced from WordNet [14] or large language models (LLMs) [60, 61]. The predictor selects the most relevant unseen class by identifying the nearest leaf node to ensure a minimal distribution distance to seen classes. The caption-based domain information generator extracts domain attributes, such as styles and backgrounds, from the seen-class data via image captioning. The text-to-image model (*e.g.*, Stable Diffusion [50]) is utilized to generate unseen-class data, with the guidance of the generated domain information and predicted unseen classes. Supported by our theoretical guarantees, this process significantly reduces the distribution gap, enabling accurate distribution estimation in open environments.

With the generated data, the distribution alignment algorithm is proposed to estimate and maximize the posterior probability of model outputs in open environments. We derive an evidence lower bound (ELBO) of the logarithmic posterior, which can be approximated as the expectation of logarithmic posterior on seen-class data minus the Kullback–Leibler (KL) divergence between the distributions of seen-class and generated unseen-class data. Hence, we employ a KL-based loss to minimize the distribution gap between the seen-class and generated unseen-class data. However, due to inherent variations in mini-batches, enforcing strict alignment in every iteration introduces misalignment, thereby degrading learning performance. To address this, we propose a sparse loss computation strategy that accumulates output distributions across iterations and then minimizes the alignment loss periodically. This approach effectively mitigates misalignment while ensuring that the posterior probability is maximized, thereby enhancing the generalization capability in open environments.

We evaluate the proposed method in open environments across two settings: base-to-base/base-to-new, and cross-dataset, using 11 image recognition datasets. Our method consistently outperforms the baseline on all datasets across two settings, demonstrating its effectiveness. Notably, on the EuroSAT dataset, our method achieves significant improvements of 14% and 9.48% in the base-to-new and cross-dataset settings, respectively, highlighting its effectiveness and superiority. Furthermore, ablation studies confirm the impact of the proposed class-domain-wise data generation pipeline and the distribution alignment algorithm. The results reveal that reducing the distribution distance between seen-class and generated unseen-class data indeed enhances performance in open environments, confirming the critical role of the proposed method. The contributions can be summarized as:

- We present a theoretical analysis that demonstrates the distributions in open environments can be effectively approximated by generating unseen-class data, with an upper bound on estimation error.

- We propose a novel open-vocabulary learning method, which introduces a class-domain-wise data generation pipeline to generate unseen-class data and a distribution alignment algorithm to accurately estimate and utilize the distribution in open environments for enhancing performance in open environments.

## 2 Related Work

### 2.1 Open-Vocabulary Learning

Existing open-vocabulary learning methods can be categorized into pre-training and prompt learning. Pre-training methods, such as CLIP [48] and ALIGN [22], train vision-language models (VLMs) on large-scale image-text pairs (*e.g.*, 400M to 1B) to learn rich multi-modal representations. Recent efforts focus on scaling datasets (*e.g.*, Datacomp [15], LAION-5B [56]) or improving training strategies (*e.g.*, caption diversity enhancement [34], fine-grained semantic alignment [36], masked cross-modal learning [58], scalable training optimization [64], multimodal guidance alignment [31] ).

However, these methods often require retraining from scratch, which is resource-intensive in terms of time, data, and annotations.

Prompt learning methods address the retraining issue by introducing learnable prompt tokens at the input [17]. Initially successful in NLP tasks [32, 35, 40], these methods are adapted for vision-language models (VLMs). CoOp pioneers continuous prompt optimization in the language branch [78], while CoCoOp improves generalization by generating conditional prompts based on visual features [77]. VPT extends this to the visual branch by optimizing visual prompt tokens [23]. Recent advancements include multi-modal prompt fusion [26, 74, 70], regularization techniques [79, 33, 27, 44], and leveraging local VLM features to enhance performance [30, 59, 6]. These methods focus on estimating the open-environment distribution using seen-class data without theoretical guarantees for upper-bounded estimation error. In contrast to existing methods that rely solely on seen-class data, we explore learning beyond the seen by generating unseen-class data for accurate and bounded distribution estimation in open environments.

## 2.2 Learning from Synthetic Data

Synthetic data enhances performance across computer vision tasks like object detection [47, 53], semantic segmentation [8, 52], autonomous driving [1], and robotics [42, 73]. Recent text-to-image models, powered by diffusion techniques, generate high-quality images from text [54, 4, 50]. Existing methods combine descriptive prompts and classes to create synthetic images [12], which, when paired with real data, improve tasks like image classification [2, 11, 18, 38, 55, 62], object detection [7, 68, 76], and semantic segmentation [16, 28, 46, 66, 67, 71]. Unlike these methods that focus on generation of seen classes, our method generates images for unseen classes to precisely estimate the open-environment distribution.

# 3 Preliminaries

## 3.1 CLIP

We implement our method on CLIP [48] that consists of an image encoder $f_{\boldsymbol{\Phi}_1}(\cdot)$ and a text encoder $g_{\boldsymbol{\Phi}_2}(\cdot)$, where parameters are $\boldsymbol{\Phi} = [\boldsymbol{\Phi}_1, \boldsymbol{\Phi}_2]$. Given an image $\boldsymbol{x}$, the image encoder embeds $\boldsymbol{x}$ and adds a learnable class token to obtain the visual feature. Then, the text encoder projects the corresponding class label $\boldsymbol{y}$ wrapped within a text template to get the textual feature. Given image $\boldsymbol{x}$ and a ground-truth $\bar{\boldsymbol{y}}$ from all classes, CLIP computes the posterior probability as

$$p(\bar{\boldsymbol{y}}|\boldsymbol{x}, \boldsymbol{\Phi}) = \frac{\exp(\text{sim}(g_{\boldsymbol{\Phi}_2}(\bar{\boldsymbol{y}}), f_{\boldsymbol{\Phi}_1}(\boldsymbol{x}))/\tau)}{\sum_i \exp(\text{sim}(g_{\boldsymbol{\Phi}_2}(\boldsymbol{y}_i), f_{\boldsymbol{\Phi}_1}(\boldsymbol{x})/\tau)}, \tag{1}$$

where $\text{sim}(\cdot, \cdot)$ is the cosine similarity and $\tau$ is a temperature parameter. In this paper, we utilize the prompt learning to optimize CLIP. We add learnable language and visual prompts given as $\boldsymbol{v}_1$ and $\boldsymbol{v}_2$ to the textual and visual inputs, respectively. Prompts $\boldsymbol{v} = [\boldsymbol{v}_1, \boldsymbol{v}_2]$ are optimized with the loss as

$$\mathcal{L}_{\text{CE}} = -\log \frac{\exp(\text{sim}(\tilde{f}_p, \tilde{g}_{p\bar{y}})/\tau)}{\sum_i \exp(\text{sim}(\tilde{f}_p, \tilde{g}_{py_i})/\tau)}, \tag{2}$$

where $\tilde{f}_p = f_{\boldsymbol{\Phi}_1}([\boldsymbol{v}_1, \boldsymbol{x}])$, and $\tilde{g}_{py_i} = g_{\boldsymbol{\Phi}_2}([\boldsymbol{v}_2, \boldsymbol{y}_i])$.

## 3.2 Open-Vocabulary Learning

Open-vocabulary learning requires recognizing objects of unseen classes and seen classes in open environments. We denote the data in open environments as $\mathcal{D}_o$, which consists of image-label pairs $\{(\boldsymbol{x}_o, \boldsymbol{y}_o)\}$. Accordingly, $\mathcal{D}_o$ can be divided into two disjoint datasets, *i.e.*, the seen-class dataset $\mathcal{D}_s = \{(\boldsymbol{x}_s, \boldsymbol{y}_s)\}$ and the unseen-class dataset $\mathcal{D}_u = \{(\boldsymbol{x}_u, \boldsymbol{y}_u)\}$. The corresponding label sets are denoted as $\boldsymbol{Y}_o = \{\boldsymbol{y}_o\}, \boldsymbol{Y}_s = \{\boldsymbol{y}_s\}, \boldsymbol{Y}_u = \{\boldsymbol{y}_u\}$, which satisfy that $\boldsymbol{Y}_u \cap \boldsymbol{Y}_s = \emptyset$ and $\boldsymbol{Y}_u \cup \boldsymbol{Y}_s = \boldsymbol{Y}_o$. A unique aspect of open-vocabulary recognition tasks is the inclusion of language vocabulary knowledge encoded in a large vocabulary space, such as the description of textual classes.

Open-vocabulary learning tasks aim to maximize the model outputs of posterior $p(\bar{\boldsymbol{y}}|\boldsymbol{x}_o, \boldsymbol{\Phi})$. Intuitively, the posterior distribution $p(\bar{\boldsymbol{y}}|\boldsymbol{x}_o, \boldsymbol{\Phi})$ is strong related to $p(\bar{\boldsymbol{y}}|\boldsymbol{x}_s, \boldsymbol{\Phi})$ and $p(\bar{\boldsymbol{y}}|\boldsymbol{x}_u, \boldsymbol{\Phi})$. Existing

methods tend to utilize seen-class data to model $p(\bar{y}|\boldsymbol{x}_s, \boldsymbol{\Phi})$, which is further to estimate $p(\bar{y}|\boldsymbol{x}_o, \Phi)$. Ignoring $p(\bar{y}|\boldsymbol{x}_u, \boldsymbol{\Phi})$ results in significant estimation errors that cannot be guaranteed to be bounded, further impacting the generalization in open environments.

## 4 Theoretical Analysis

To facilitate open-vocabulary learning, we explore learning beyond the seen classes for accurate distribution estimation in open environments. In this section, we demonstrate that the distribution in open environments can be estimated by generating unseen-class data $\mathcal{G}_u = \{(\boldsymbol{x}_e, \boldsymbol{y}_e)\}$. The label set of $\mathcal{G}_u$ is denoted as $\boldsymbol{Y}_e = \{\boldsymbol{y}_e\}$ that satisfies $\boldsymbol{Y}_e \cap \boldsymbol{Y}_s = \emptyset$ and $\boldsymbol{Y}_e \cup \boldsymbol{Y}_s = \boldsymbol{Y}_o$. Our theoretical analysis is conducted from two perspectives, *i.e.*, the joint probability distribution and the posterior probability distribution.

The joint probability distribution in open environments is denoted as $p(\boldsymbol{x}_o, \boldsymbol{y}_o)$. Due to $\boldsymbol{Y}_e \cap \boldsymbol{Y}_s = \emptyset$ and $\boldsymbol{Y}_u \cup \boldsymbol{Y}_s = \boldsymbol{Y}_o$, $p(\boldsymbol{x}_o, \boldsymbol{y}_o)$ can be modeled as

$$p(\boldsymbol{x}_o, \boldsymbol{y}_o) = p(\boldsymbol{x}_s, \boldsymbol{y}_s) + p(\boldsymbol{x}_u, \boldsymbol{y}_u), \tag{3}$$

where $p(\boldsymbol{x}_s, \boldsymbol{y}_s)$ and $p(\boldsymbol{x}_u, \boldsymbol{y}_u)$ denote the joint probability distribution of seen classes and unseen classes, and $p(\boldsymbol{x}_s, \boldsymbol{y}_s)$ can be directly modeled from seen-class data. We propose that $p(\boldsymbol{x}_u, \boldsymbol{y}_u)$ can be estimated by generating unseen-class data $\mathcal{G}_u = \{(\boldsymbol{x}_e, \boldsymbol{y}_e)\}$, where this estimation error is upper bounded, as shown in Theorem 1.

**Theorem 1.** *With probability at least $1 - \delta$, we have the following,*

$$d\big(p(\boldsymbol{x}_e, \boldsymbol{y}_e), p(\boldsymbol{x}_u, \boldsymbol{y}_u)\big) \leq \sqrt{\frac{d\big(p(\boldsymbol{x}_e, \boldsymbol{y}_e), p(\boldsymbol{x}_s, \boldsymbol{y}_s)\big)}{2m - 1}} + \sqrt{\frac{\ln\frac{1}{\delta} + \frac{5}{2}\ln m + 8}{2m - 1}}, \tag{4}$$

*where $d(\cdot, \cdot)$ denotes the distribution distance, and $m$ denotes the size of seen-class dataset.*

This theorem demonstrates that the distance between the joint distributions of unseen-class and generated unseen-class data has an upper bound. In Theorem 1, we can observe that as the distance between the joint distributions of generated unseen-class data and seen-class data $d\big(p(\boldsymbol{x}_e, \boldsymbol{y}_e), p(\boldsymbol{x}_s, \boldsymbol{y}_s)\big)$ decreases, the upper bound in Eq. (4) decreases. This indicates that we can narrow the gap between the joint distributions of generated unseen-class data and unseen-class data by reducing the distance between the joint distributions of generated unseen-class data and seen-class data. Theorem 1 holds for any distribution distance (*e,g,*, KL divergence, total variation distance, or other forms).

In terms of posterior probability distribution, we demonstrate that the estimation error between $p(\bar{y}|\boldsymbol{x}_e, \boldsymbol{\Phi})$ and $p(\bar{y}|\boldsymbol{x}_u, \boldsymbol{\Phi})$ is upper bounded, which is presented in Theorem 2. Without loss of generality, we define the distribution distance as KL divergence for analysis.

**Theorem 2.** *Given the predicted classes $\boldsymbol{Y}_e = \{\boldsymbol{y}_e\}$. Suppose that predicted class $\boldsymbol{Y}_e$ have any nonzero probability $p(\boldsymbol{Y}_e)$. With probability at least $1 - \delta$ over the $m$ instances of generated unseen-class data $\{(\boldsymbol{x}_e, \boldsymbol{y}_e)\}$, we have that*

$$D_{KL}\big(p(\bar{y}|\boldsymbol{x}_u, \boldsymbol{\Phi})||p(\bar{y}|\boldsymbol{x}_e, \boldsymbol{\Phi})\big) \leq \sqrt{\frac{\ln\frac{1}{p(\boldsymbol{Y}_e)} + \ln\frac{1}{\delta}}{2m}} - \ln\frac{|\boldsymbol{Y}_e|}{|\boldsymbol{Y}_u|}. \tag{5}$$

**Discussion.** $p(\boldsymbol{Y}_e)$ is the probability assigned to $\boldsymbol{Y}_e$ under the distribution over $\boldsymbol{Y}_u$. Since $\boldsymbol{Y}_e \subset \boldsymbol{Y}_u$, $p(\boldsymbol{Y}_e) > 0$ always holds without any constraint on $\boldsymbol{Y}_e$.

Theorem 2 demonstrate that the estimation of the posterior probability distribution in open environments has an upper bound. From Theorem 2, we can observe that $\lim_{m\to\infty} \sqrt{\frac{\ln\frac{1}{p(\boldsymbol{Y}_e)} + \ln\frac{1}{\delta}}{2m}} = 0$, which indicates that increasing the amount of samples $m$ can decrease the approximation error. As the probability $p(\boldsymbol{Y}_e)$ and $|\boldsymbol{Y}_e|$ increase, the approximation error bound decreases, which conforms to common sense. The step-by-step derivations are presented in the appendix.

From Theorems 1 and 2, we can conclude that the distance between the distributions of generated unseen-class data and unseen-class data in open environments has an upper bound. This provides the theoretical guarantee for the open-environment distribution estimation.

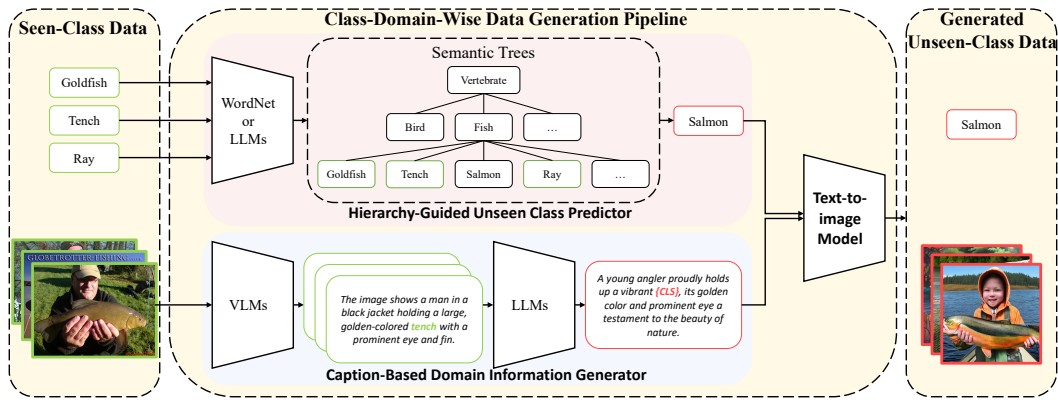

Figure 1: Formulation of Class-Domain-Wise Data Generation Pipeline

# 5 Method

Based on the theoretical analysis, we propose a novel open-vocabulary learning method that includes a class-domain-wise data generation pipeline to generate unseen-class data and a distribution alignment algorithm to estimate and utilize the estimated distribution for generalization.

## 5.1 Class-Domain-Wise Data Generation Pipeline

Inspired by Theorem 1 that indicates estimation error is related to the distribution distance between seen-class data and generated unseen-class data, our goal is to generate unseen-class data aligned with the seen-class data distribution. Our pipeline includes a **hierarchy-guided unseen class predictor** to identify classes close to seen-class data, a **caption-based domain information generator** to extract domain information of seen-class data, and a **text-to-image model**. These components work together to generate unseen-class data that align with the seen-class data distribution. The overall pipeline is shown in Figure 1.

### 5.1.1 Hierarchy-Guided Unseen Class Predictor

In open environments, the semantic structure of classes exhibits a prominent hierarchical nature, akin to the label space in ImageNet [10], which is also organized hierarchically. Motivated by this observation, the unseen class predictor leverages the semantic structure of seen classes to predict unseen classes.

We first construct a semantic tree for the seen classes $\boldsymbol{Y}_s$, where $\boldsymbol{Y}_s$ are the leaf nodes and parent nodes represent superclasses derived from WordNet [14] or large language models (LLMs) [60, 61]. If WordNet contains the classes, superclasses are their hypernyms; otherwise, LLMs are queried. To predict unseen classes, we expand the semantic tree by adding missing hyponyms of the superclasses via WordNet or LLMs. These hyponyms, representing sibling nodes of the seen classes, serve as potential unseen class candidates. For each candidate, we compute its cosine similarity with seen classes using textual embeddings from text encoder $g_{\boldsymbol{\Phi}_2}(\cdot)$, and select the top $K_0$ closest candidates as predicted unseen classes.

### 5.1.2 Caption-Based Domain Information Generator

Domain information (such as image style and scene information) plays important roles in the alignment of generated unseen-class data and seen-class data. We aim to capture the domain information from seen-class data by utilizing Vision-Language Models (VLMs) [39]. In doing so, we have to encounter two main issues. Firstly, the hallucination originated from VLMs may result in unmatched domain information. Secondly, domain information extracted from seen-class data may be limited, undermining the diversity of generated images.

To solve these issues, we utilize VLMs to generate class-specific captions for each classes, and then we calculate the similarity between these descriptions and the images of corresponding classes. By

selecting top $K_1$ results with the highest similarity, we ensure that the generated textual descriptions closely align with the image content, thereby effectively reducing hallucination issues. Then, the selected captions are summarized as top $K_2$ class-specific domain information by using LLMs.

### 5.1.3 Text-to-Image Model

The text-to-image model [51] is utilized to generate unseen-class data, with the guidance of the predicted unseen classes and the corresponding class-specific domain information.

Because the unseen classes are close to seen classes and domain information aligns with seen-class data, we narrow the distribution gap between the generated unseen-class data and the seen-class data.

## 5.2 Distribution Alignment

After generating unseen-class data $\mathcal{G}_u = \{(\boldsymbol{x}_e, \boldsymbol{y}_e)\}$, we estimate and maximize the posterior probability in open environments, thereby enhancing its generalization. The posterior probability distribution of model outputs in open environments satisfies that

$$p(\bar{\boldsymbol{y}}|\boldsymbol{x}_o, \boldsymbol{\Phi}) \propto p(\bar{\boldsymbol{y}}|\boldsymbol{x}_u, \boldsymbol{\Phi})p(\bar{\boldsymbol{y}}|\boldsymbol{x}_s, \boldsymbol{\Phi}). \tag{6}$$

The Evidence Lower Bound (ELBO) of the logarithmic posterior probability can be derived as

$$\log p(\bar{\boldsymbol{y}}|\boldsymbol{x}_o, \boldsymbol{\Phi}) \geq \mathbb{E}[\log p(\bar{\boldsymbol{y}}|\boldsymbol{x}_s, \boldsymbol{\Phi})] - D_{\mathrm{KL}}(p(\bar{\boldsymbol{y}}|\boldsymbol{x}_s, \boldsymbol{\Phi})||p(\bar{\boldsymbol{y}}|\boldsymbol{x}_u, \boldsymbol{\Phi})), \tag{7}$$

where the proof is presented in the appendix. Due to the absence of $(\boldsymbol{x}_u, \boldsymbol{y}_u)$, we leverage the generated unseen-class data $(\boldsymbol{x}_e, \boldsymbol{y}_e)$ to estimate ELBO, *i.e.*,

$$\mathbb{E}[\log p(\bar{\boldsymbol{y}}|\boldsymbol{x}_s, \boldsymbol{\Phi})] - D_{\mathrm{KL}}(p(\bar{\boldsymbol{y}}|\boldsymbol{x}_s, \boldsymbol{\Phi})||p(\bar{\boldsymbol{y}}|\boldsymbol{x}_e, \boldsymbol{\Phi})). \tag{8}$$

As Theorem 2 suggests that the KL divergence between $p(\bar{\boldsymbol{y}}|\boldsymbol{x}_u, \boldsymbol{\Phi})$ and $p(\bar{\boldsymbol{y}}|\boldsymbol{x}_e, \boldsymbol{\Phi})$ is upper-bounded, this estimation in Eq. (8) is reasonable and practically effective. Thus, $\log p(\bar{\boldsymbol{y}}|\boldsymbol{x}_o, \boldsymbol{\Phi})$ in Eq. (54) can be maximized by minimizing $-\mathbb{E}[\log p(\bar{\boldsymbol{y}}|\boldsymbol{x}_s, \boldsymbol{\Phi})]$ and $D_{\mathrm{KL}}(p(\bar{\boldsymbol{y}}|\boldsymbol{x}_s, \boldsymbol{\Phi})||p(\bar{\boldsymbol{y}}|\boldsymbol{x}_e, \boldsymbol{\Phi}))$.

To this end, we design distribution alignment algorithm that adopts prompt learning to optimize CLIP. Specifically, we minimize the $\mathcal{L}_{\mathrm{CE}}(\cdot)$ on $\mathcal{D}_b$ in Eq. (2) to minimize $-\mathbb{E}[\log p(\bar{\boldsymbol{y}}|\boldsymbol{x}_s, \boldsymbol{\Phi})]$. To minimize $D_{\mathrm{KL}}(p(\bar{\boldsymbol{y}}|\boldsymbol{x}_s, \boldsymbol{\Phi})||p(\bar{\boldsymbol{y}}|\boldsymbol{x}_e, \boldsymbol{\Phi}))$, we introduce a KL-based loss for distribution alignment, which is formulated as

$$\mathcal{L}_{\mathrm{KL}} = D_{\mathrm{KL}}\big[p(\bar{\boldsymbol{y}}|\boldsymbol{x}_b, \boldsymbol{\Phi}, \boldsymbol{v})||p(\bar{\boldsymbol{y}}|\boldsymbol{x}_e, \boldsymbol{\Phi}, \boldsymbol{v})\big]. \tag{9}$$

By minimizing $\mathcal{L}_{\mathrm{CE}}(\cdot)$ and $\mathcal{L}_{\mathrm{KL}}(\cdot)$, $p(\bar{\boldsymbol{y}}|\boldsymbol{x}_o, \boldsymbol{\Phi})$ is maximized.

The proposed loss $\mathcal{L}_{\mathrm{KL}}(\cdot)$ introduces additional challenges, *i.e.*, the data from unseen and seen classes in the mini-batch may differ significantly, and the loss could forcefully align their output distributions despite their large inherent differences. This misalignment between the distributions can, in turn, compromise the learning performance of both the base and unseen classes. To address the issue, we propose a sparse loss computation strategy that accumulates output distributions of seen-class data across iterations and then minimizes the alignment loss periodically. During each iteration, we save the output distributions of the seen-class data. For each batch of unseen-class data, we compute the similarity between the saved output distributions of seen-class data and that of the unseen-class data. Alignment is then performed on the top $K_3$ most similar distributions, which helps alleviate the misalignment problem by ensuring a more accurate and consistent alignment across batches.

In order to decrease the distribution distance between the generated data and real data in the feature spaces, we introduce a Maximum Mean Discrepancy (MMD) loss. Specifically, we first generate some extra seen-class data $\mathcal{G}_s = \{(\boldsymbol{x'}_s, \boldsymbol{y}_s)\}$ using the same generation pipeline as presented in Section 5.1. Then the MMD loss is formulated as

$$\mathcal{L}_{\mathrm{MMD}} = \frac{1}{n^2} \sum_{i,j=1}^{n} K(\boldsymbol{x}_s^i, \boldsymbol{x}_s^j) + \frac{1}{n^2} \sum_{i,j=1}^{n} K(\boldsymbol{x'}_b^i, \boldsymbol{x'}_b^j) - \frac{2}{n^2} \sum_{i=1}^{n} \sum_{j=1}^{n} K(\boldsymbol{x}_i, \boldsymbol{x'}_j), \tag{10}$$

where $n$ is batch size. $K(\boldsymbol{x}, \boldsymbol{y}) = e^{-\frac{\|\boldsymbol{x}-\boldsymbol{y}\|^2}{2\sigma^2}}$ represents Gaussian kernel. By minimizing the MMD loss, the distance of feature spaces between generated data and real data is decreased, further improving alignment. We also employ the sparse loss computation strategy for the MMD loss.

Overall, the loss function for updating parameters is

$$\mathcal{L}_{\mathrm{total}} = \mathcal{L}_{\mathrm{CE}} + \alpha \mathcal{L}_{\mathrm{KL}} + \beta \mathcal{L}_{\mathrm{MMD}}, \tag{11}$$

where $\alpha$ and $\beta$ are hyper-parameters. By minimizing $\mathcal{L}_{\text{total}}$, the posterior probability in open environments $p(\bar{\boldsymbol{y}}|\boldsymbol{x}_o, \boldsymbol{\Phi})$ is maximized, thereby improving capability in open environments. This algorithm is summarized in appendix.

Table 1: Results in base-to-new/base-to-base generalization setting. We bold the best results and underline the second-best results. H denotes the harmonic mean of performance on base and new.

| Dataset | | CLIP [48] | CoOp [78] | CoCoOp [77] | DePT [75] | TCP [72] | CuTCP [21] | DeKg [37] | PromptSRC (baseline) | Ours | Gain Δ |
|---|---|---|---|---|---|---|---|---|---|---|---|
| Average | Base | 69.34 | 82.69 | 80.47 | 85.19 | 84.13 | 84.21 | 84.96 | 84.26 | **86.40** | +2.14 |
| | New | 74.22 | 63.22 | 71.69 | 76.17 | 75.36 | 76.10 | 76.38 | 76.10 | **80.52** | +4.42 |
| | H | 71.70 | 71.66 | 75.83 | 80.43 | 79.51 | 79.95 | 80.44 | 79.97 | **83.36** | +3.39 |
| ImageNet | Base | 72.43 | 76.47 | 75.98 | **78.20** | 77.27 | 77.73 | 77.40 | 77.60 | 77.91 | +0.31 |
| | New | 68.14 | 67.88 | 70.43 | 70.27 | 69.87 | 70.50 | 69.20 | 70.73 | **70.74** | +0.01 |
| | H | 70.22 | 71.92 | 73.10 | 74.02 | 73.38 | 73.94 | 73.07 | 74.01 | **74.15** | +0.14 |
| Caltech | Base | 96.84 | 98.00 | 97.96 | 98.57 | 98.23 | 98.47 | 98.64 | 98.10 | **98.97** | +0.87 |
| | New | 94.00 | 89.81 | 93.81 | 94.10 | 94.67 | 95.27 | 95.20 | 94.03 | **95.85** | +1.82 |
| | H | 95.40 | 93.73 | 95.84 | 96.28 | 96.42 | 96.84 | 96.89 | 96.02 | **97.38** | +1.36 |
| Pets | Base | 91.17 | 93.67 | 95.20 | 95.43 | 94.67 | 95.07 | 94.47 | 95.33 | **96.01** | +0.68 |
| | New | 97.26 | 95.29 | 97.69 | 97.33 | 97.20 | 97.83 | 97.76 | 97.30 | **98.27** | +0.97 |
| | H | 94.12 | 94.47 | 96.43 | 96.37 | 95.92 | 96.43 | 96.09 | 96.30 | **97.12** | +0.82 |
| Cars | Base | 63.37 | 78.12 | 70.49 | 80.80 | 80.80 | 80.23 | 81.18 | 78.27 | **82.93** | +4.66 |
| | New | 74.89 | 60.40 | 73.59 | 75.00 | 74.13 | 74.27 | 74.75 | 74.97 | **80.81** | +5.84 |
| | H | 68.65 | 68.13 | 72.01 | 77.79 | 77.32 | 77.13 | 77.83 | 76.58 | **81.86** | +5.28 |
| Flowers | Base | 72.08 | 97.60 | 94.87 | 98.40 | 97.73 | 98.10 | 98.58 | 98.07 | **98.77** | +0.70 |
| | New | 77.80 | 59.67 | 71.75 | 77.10 | 75.57 | 75.58 | 75.18 | 76.50 | **80.92** | +4.42 |
| | H | 74.83 | 74.06 | 81.71 | 86.46 | 85.23 | 85.38 | 85.30 | 85.95 | **88.96** | +3.01 |
| Food | Base | 90.10 | 88.33 | 90.70 | 90.87 | 90.57 | 90.47 | 90.73 | 90.67 | **91.39** | +0.72 |
| | New | 91.22 | 82.26 | 91.29 | 91.57 | 91.37 | 91.77 | 91.55 | 91.53 | **92.99** | +1.46 |
| | H | 90.66 | 85.19 | 90.99 | 91.22 | 90.97 | 91.11 | 91.14 | 91.10 | **92.18** | +1.08 |
| Aircraft | Base | 27.19 | 40.44 | 33.41 | 45.70 | 41.97 | 42.43 | 45.20 | 42.73 | **48.98** | +6.25 |
| | New | 36.29 | 22.30 | 23.71 | 36.73 | 34.43 | 36.37 | 35.09 | 37.87 | **44.03** | +6.16 |
| | H | 31.09 | 28.75 | 27.74 | 40.73 | 37.83 | 39.17 | 39.51 | 40.15 | **46.37** | +6.22 |
| SUN | Base | 69.36 | 80.60 | 79.74 | 83.27 | 82.63 | 83.00 | 82.52 | 82.67 | **83.64** | +0.97 |
| | New | 75.35 | 65.89 | 76.86 | 78.97 | 78.20 | 78.23 | 78.30 | 78.47 | **80.15** | +1.68 |
| | H | 72.23 | 72.51 | 78.27 | 81.06 | 80.35 | 80.55 | 80.35 | 80.52 | **81.86** | +1.34 |
| DTD | Base | 53.24 | 79.44 | 77.01 | 84.80 | 82.77 | 83.00 | 83.80 | 83.37 | **85.53** | +2.16 |
| | New | 59.90 | 41.18 | 56.00 | 61.20 | 58.07 | 59.40 | 59.66 | 62.97 | **71.50** | +8.53 |
| | H | 56.37 | 54.24 | 64.85 | 71.09 | 68.25 | 69.24 | 69.70 | 71.75 | **77.89** | +6.14 |
| EuroSAT | Base | 56.48 | 92.19 | 87.49 | 93.23 | 91.63 | 90.87 | 94.02 | 92.90 | **97.17** | +4.27 |
| | New | 64.05 | 54.74 | 60.04 | 77.90 | 74.73 | 77.13 | 81.69 | 73.90 | **87.90** | +14.00 |
| | H | 60.03 | 68.69 | 71.21 | 84.88 | 82.32 | 83.44 | 87.42 | 82.32 | **92.30** | +9.98 |
| UCF | Base | 70.53 | 84.69 | 82.33 | 87.73 | 87.13 | 86.87 | 88.06 | 87.10 | **89.14** | +2.04 |
| | New | 77.50 | 56.05 | 73.45 | 77.70 | 80.77 | 80.80 | 81.77 | 78.80 | **82.53** | +3.73 |
| | H | 73.85 | 67.46 | 77.64 | 82.46 | 83.83 | 83.72 | 84.80 | 82.74 | **85.71** | +2.97 |

# 6 Experiments

## 6.1 Experiment Settings

We evaluate our method on open-vocabulary benchmarks with 11 image recognition datasets, following the setting of the baseline method PromptSRC [27].

**Datasets**. We evaluate the proposed method on 11 image recognition datasets: ImageNet [10], Caltech101 (Caltech) [13], OxfordPets (Pets) [45], StanfordCars (Cars) [29], Flowers102 (Flowers) [43], Food101 (Food) [5], FGVCAircraft (Aircraft) [41], SUN397 (SUN) [69], UCF101 (UCF) [57], DTD [9] and EuroSAT [19].

**Benchmark Settings.** We evaluate our method on two open-vocabulary learning benchmarks.

Table 2: Results of our method and state-of-the-art methods for cross-dataset evaluation.

| | Source | Target | | | | | | | | | | |
|---|---|---|---|---|---|---|---|---|---|---|---|---|
| | ImageNet | Caltech | Pets | Cars | Flowers | Food | Aircraft | SUN | DTD | EuroSAT | UCF | Average |
| CoOp | **71.51** | 93.70 | 89.14 | 64.51 | 68.71 | 85.30 | 18.47 | 64.15 | 41.92 | 46.39 | 66.55 | 63.88 |
| CoCoOp | 71.02 | 94.43 | 90.14 | 65.32 | 71.88 | 86.06 | 22.94 | 67.36 | 45.73 | 45.37 | 68.21 | 65.74 |
| ASPrompt | 71.05 | **94.57** | **90.79** | 66.90 | 72.30 | 86.17 | **25.16** | 67.32 | 47.35 | 50.25 | 69.52 | 67.03 |
| PromptSRC | 71.27 | 93.60 | 90.25 | 65.70 | 70.25 | 86.15 | 23.90 | 67.10 | 46.87 | 45.50 | 68.75 | 65.81 |
| Ours | 71.22 | 93.87 | 90.46 | **67.36** | **72.88** | **86.61** | 25.14 | **67.68** | **48.27** | **54.98** | **69.44** | **67.68** |
| Gain Δ | -0.05 | +0.27 | +0.21 | +1.66 | +2.63 | +0.46 | +1.24 | +0.58 | +1.40 | +9.48 | +0.72 | +1.87 |

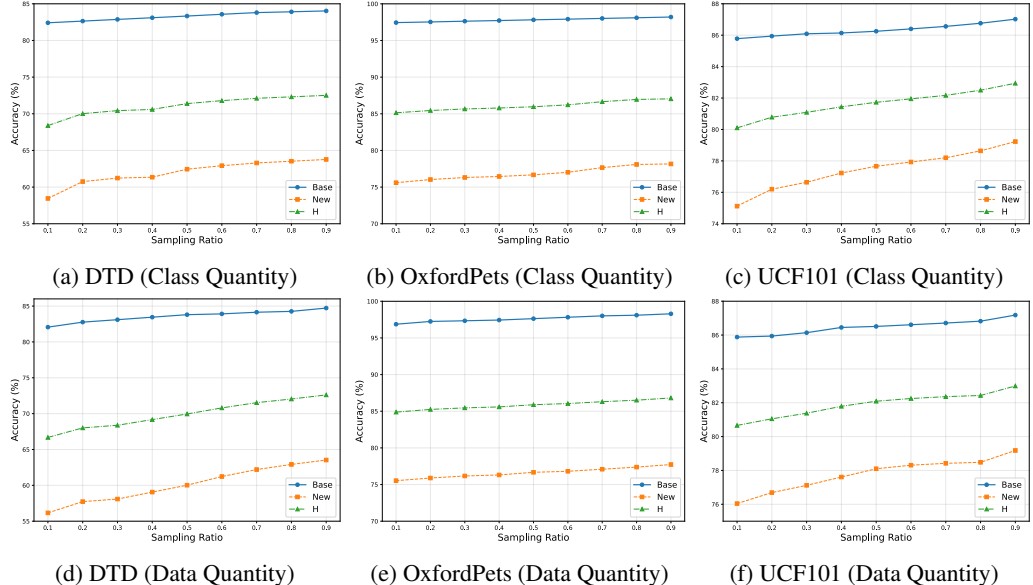

(a) DTD (Class Quantity)    (b) OxfordPets (Class Quantity)    (c) UCF101 (Class Quantity)

(d) DTD (Data Quantity)    (e) OxfordPets (Data Quantity)    (f) UCF101 (Data Quantity)

Figure 2: Ablation Studies on Quantity of Predicted Unseen Classes and Generated Images.

- **Base-to-base/Base-to-new generalization.** We equally split each dataset into base and new classes. The model is trained on base classes and evaluated on both base classes (base-to-base) and new classes (base-to-new) across all 11 datasets.

- **Cross-dataset evaluation.** To evaluate our method on the unseen classes under different domain environments, we adopt cross-dataset setting, where we train a model on ImageNet (source domain) and evaluate the model on the other 10 datasets (target domains) without any fine-tuning.

**Implementation details**. Following the setting of PromptSRC [27], we use a few-shot training strategy in all experiments at 16 shots which are randomly sampled for each class. We apply prompt learning on a pretrained ViT-B/16 based CLIP model and report results averaged over 3 runs. Other details such as hyper-parameters are provided in appendix.

### 6.2 Base-to-Base/Base-to-New Generalization

We compare our method with other prompt learning methods in Table 1. Results show that our method significantly improves the performance of PromptSRC in open environments for both base-to-base and base-to-new settings, demonstrating its effectiveness in open environments. Notably, on new classes, our method achieves up to a $14\%$ performance improvement, indicating its ability to effectively estimate the distribution of unseen classes.

We also compare our method with other state-of-the-art methods: CLIP [48], CoOp [78], Co-CoOp [77], DePT [75], TCP [72], CuTCP [21], and DeKg [37]. Results show that our method

Table 3: Ablation studies on class quality and data quality. "Acc" denotes the accuracy. "Dis" denotes the distribution distance of the generated unseen-class data and seen-class data.

| Dataset | | Ours | | Class Quality | | | | Data Quality | | | | | |
|---|---|---|---|---|---|---|---|---|---|---|---|---|---|
| | | | | LowSim | | w/o Tree | | Picture | | Photo | | Image | |
| | | Acc ↑ | Dis ↓ | Acc ↑ | Dis ↓ | Acc ↑ | Dis ↓ | Acc ↑ | Dis ↓ | Acc ↑ | Dis ↓ | Acc ↑ | Dis ↓ |
| Caltech | Base | **98.97** | | 98.26 | | 97.93 | | 98.26 | | 98.19 | | 98.06 | |
| | New | **95.85** | **9.99** | 94.32 | 10.49 | 93.67 | 13.16 | 94.21 | 11.84 | 94.11 | 11.95 | 93.78 | 12.12 |
| | H | **97.38** | | 96.25 | | 95.75 | | 96.19 | | 96.11 | | 95.87 | |
| Pets | Base | **96.01** | | 95.85 | | 95.00 | | 95.69 | | 95.43 | | 95.27 | |
| | New | **98.27** | **7.58** | 97.60 | 8.19 | 96.81 | 11.71 | 97.04 | 9.14 | 96.98 | 9.21 | 96.92 | 10.27 |
| | H | **97.12** | | 96.72 | | 95.90 | | 96.36 | | 96.20 | | 96.09 | |
| Cars | Base | **82.93** | | 78.94 | | 77.86 | | 77.99 | | 78.71 | | 78.24 | |
| | New | **80.81** | **8.92** | 75.29 | 9.33 | 73.36 | 13.78 | 74.75 | 10.65 | 75.04 | 10.08 | 74.92 | 10.48 |
| | H | **81.86** | | 77.07 | | 75.54 | | 76.33 | | 76.83 | | 76.54 | |
| Flowers | Base | **98.77** | | 98.29 | | 97.15 | | 97.82 | | 97.91 | | 97.63 | |
| | New | **80.92** | **6.29** | 77.52 | 6.88 | 75.04 | 10.29 | 76.88 | 7.99 | 77.09 | 7.84 | 76.17 | 8.22 |
| | H | **88.96** | | 86.68 | | 84.67 | | 86.09 | | 86.26 | | 85.57 | |

Table 4: Ablation study on distribution alignment. "w/o da" denotes prompt learning without distribution alignment.

| | Caltech | | Pets | | Cars | | Flowers | | Aircraft | | DTD | | EuroSAT | | UCF | |
|---|---|---|---|---|---|---|---|---|---|---|---|---|---|---|---|---|
| | w/o da | Ours | w/o da | Ours | w/o da | Ours | w/o da | Ours | w/o da | Ours | w/o da | Ours | w/o da | Ours | w/o da | Ours |
| Base | 97.42 | **98.97** | 94.10 | **96.01** | 74.14 | **82.93** | 96.49 | **98.77** | 39.80 | **48.98** | 75.81 | **85.53** | 93.57 | **97.17** | 85.16 | **89.14** |
| New | 90.07 | **95.85** | 88.65 | **97.65** | 71.63 | **80.81** | 71.63 | **80.92** | 23.39 | **44.03** | 53.50 | **71.50** | 64.13 | **87.90** | 62.03 | **82.53** |
| H | 93.60 | **97.38** | 91.29 | **96.82** | 72.86 | **81.86** | 82.22 | **88.96** | 29.47 | **46.37** | 62.73 | **77.89** | 76.10 | **92.30** | 71.78 | **85.71** |

achieves the best performance, demonstrating its superiority in open environments, especially its generalization ability on unseen classes.

## 6.3 Cross-Dataset Evaluation

The comparison between our method and other prompt learning methods is presented in Table 2. Compared to PromptSRC, our method achieves the comparable performance on the source dataset. On the target datasets, our method significantly outperforms PromptSRC with a notable improvement of $9.48\%$ on the Eurosat dataset, demonstrating that our method can improve the generalization on unseen classes across different domains, effectively working in open-vocabulary learning task.

## 6.4 Ablation Studies

### 6.4.1 Effectiveness of Class-Domain-Wise Data Generation Pipeline

We evaluate the effectiveness of two components with respect to this pipeline, *i.e.*, hierarchy-guided unseen class predictor and caption-based domain information generator.

**Hierarchy-Guided Unseen Class Predictor.** We evaluate the effectiveness by adjusting the quantity and quality of predicted unseen classes. As to the quantity, we randomly select $s_{cls} \times N$ classes from the $N$ predicted unseen classes for training, where $s_{cls} < 1$ denotes sampling ratio. We conduct experiments on three datasets (DTD, Flowers102, UCF101), and we set $s_{cls}$ as 0.1 to 0.9 in increments of 0.1 for each dataset. As shown in Figures 2a, 2b and 2c, we observe that increases of quantity lead to better performance, especially for new classes, which is consistent with Theorem 2.

As to the quality, we introduce "LowSim" that chooses candidate classes with lowest cosine similarity and "w/o Tree" that directly ask LLMs to query unseen classes. We calculate the distribution distance between the generated unseen-class data and the seen-class data. As shown in Table 3, our method

can significantly reduce the distribution distance and improve alignment with seen-class data. Results also reveal that alignment improves recognition, which is consistent with Theorem 1.

**Caption-Based Domain Information Generator.** We evaluate its effectiveness of by adjusting the quantity and quality of generated images. As to the quantity, we randomly select $s_{imag} \times M$ images of each predicted unseen classes for training, where $s_{cls} < 1$ denotes sampling ratio and $M$ denotes the amount of data for each classes. Figures 2d, 2e and 2f show that increase of quantity leads to better performance, especially for new classes, which is consistent with Theorem 2.

As to the quality, we modify the prompt template for data generation, without the domain information inferred from the generator. We inject "Picture", "Photo" and "Image" into stable diffusion model to generate images, which are denoted as the the prompt template "A picture of a {class}", "A photo of a {class}", and "An image of a {class}", respectively. Results are shown in Table 3. We observe that our method can significantly reduce the distribution distance and improve alignment with seen-class data. Results also reveal that this alignment improves recognition, which is consistent with Theorem 1.

To further evaluate the generated image quality, we computed CLIPScore [20]. CLIPScores on UCF101, DTD, SUN397, Caltech101, OxfordPets, and StanfordCars are 0.43, 0.42, 0.43, 0.43, 0.44, and 0.42, respectively. Results show that these images exhibit high semantic quality (CLIPScore >0.35 is considered high quality [63, 3, 54]). We further conducted a user study and a GPT-4-based evaluation, where both human annotators and GPT-4 independently rated 200 randomly selected images on a 1–5 scale (5 is the highest quality). The resulting average scores of 4.67 (human) and 4.59 (GPT-4) confirm the high quality of the generated images.

### 6.4.2 Effectiveness of Distribution Alignment

We evaluate the effectiveness of the distribution alignment algorithm by directly using the generated images for prompt learning during training without the distribution alignment algorithm, denoted by "w/o pda". Results are shown in Table 5, which indicate that the proposed algorithm can improve the model performance in open environments by aligning the output distributions of model between generated data and real data.

### 6.5 Efficiency Analysis

We take Eurosat as an example for analysis. The training time and memory of our method requires at most 16.14 GB and 739.2 s. The baseline requires 6.12 GB and 101.25 s. The added time and memory mainly come from data generation. At inference time, no extra parameters or computation are introduced. Thus, our runtime (3.8 s) and memory (1.84 GB) are identical to the baseline.

## 7 Conclusion and Discussion

In this paper, we have investigated learning beyond the seen for bounded distribution estimation in the open-vocabulary task. We have demonstrated the distribution in open environments can be estimated by generating unseen-class data with upper-bounded estimation error, as evidenced by the constructed theoretical analysis. The proposed open-vocabulary learning method consists of a class-domain-wise data generation pipeline and a distribution alignment algorithm. The data generation pipeline generates unseen-class data via the introduced unseen class predictor and domain information generator, enabling accurate distribution estimation. With the generated data, the proposed distribution alignment algorithm can effectively estimate and maximize the posterior probability in open environments for improving generalization in open environments. Experiments on 11 datasets demonstrate that our method can generate unseen-class data for accurate distribution estimation, leading to consistent improvements in generalization across diverse open environments.

The generated data exists biases from the utilized LLMs and wordnet. In the future, we plan to design a multi-expert collaboration strategy that leverages diverse pretrained models and agreement-based selection to reduce dependence and bias on any single model. Moreover, to extend our method to support truly unknown classes, such as newly cartoon characters, we plan to integrate RAG mechanisms that dynamically retrieve emerging classes from external sources to enhance the ability of the model to generate up-to-date images, enabling the model to adapt in real time to new concepts.

**Acknowledgements.** This work was supported by the Shenzhen Science and Technology Program under Grant No. JCYJ20241202130548062, the Natural Science Foundation of Shenzhen under Grant No. JCYJ20230807142703006, the Natural Science Foundation of China (NSFC) under Grants No. 62406009, No. 62172041 and No. 62176021.

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

# A Proof of Theorem 3

In the manuscript, we present Theorem 3 to demonstrate that the estimation error of $p(\boldsymbol{x}_u, \boldsymbol{y}_u)$ is upper bounded. Here we provide further derivations of Theorem 3 of the manuscripts. To this end, We first review the defined setting in open-vocabulary learning task, present some lemmas, and their proofs, which are based on PAC-Bayesian Theorems [?, ?].

**Setting.** The data in open environments can be denoted as $\mathcal{D}_o = \{\mathcal{D}_s, \mathcal{D}_u\}$, which consists of image-label pairs $\{(\boldsymbol{x}_o, \boldsymbol{y}_o)\} = \{(\boldsymbol{x}_s, \boldsymbol{y}_s), (\boldsymbol{x}_u, \boldsymbol{y}_u)\}$. We assume a predicted unseen-class data distribution $\boldsymbol{E}$, where $(\boldsymbol{x}_e^i, \boldsymbol{y}_e^i)$ has the probability $\boldsymbol{E}_i$. Similarly, the distributions of training data, unseen data and data in open environments are denoted as $\boldsymbol{S}, \boldsymbol{U}$ and $\boldsymbol{O}$, respectively.

**Lemma 1.** *With probability at least $1 - \delta$ over the training dataset of size m, we have the following,*

$$\sum \boldsymbol{S}_i e^{(2m-1)\gamma_i^2} \leq \frac{4m}{\delta}. \tag{12}$$

*Proof.* By the Chernoff bound we have $P(\gamma \geq x) \leq 2e^{-2mx^2}$. We now consider the density function $f(\gamma)$ maximizing $\int_0^\infty e^{(2m-1)\gamma^2} f(\gamma) d\gamma$ subject to the constraint that $\int_x^\infty f(\gamma) d\gamma \leq 2e^{-2mx^2}$. The maximum occurs when we have $\int_x^\infty f(\gamma) d\gamma = 2e^{-2mx^2}$ which is realized when $f(\gamma) = 8m\gamma e^{-2m\gamma^2}$. So we have the following.

$$\begin{aligned}
\mathbb{E}_S \, e^{(2m-1)\gamma^2} &\leq \int_0^\infty e^{(2m-1)\gamma^2} f(\gamma) d\gamma \\
&= \int_0^\infty 8m\gamma e^{(2m-1)\gamma^2} e^{-2m\gamma^2} d\gamma \\
&= \int_0^\infty 8m\gamma e^{-\gamma^2} d\gamma \\
&= 4m,
\end{aligned} \tag{13}$$

which suffices to the following.

$$\forall i \quad \mathbb{E}_S \left[ e^{(2m-1)\gamma_i^2} \right] \leq 4m. \tag{14}$$

So we have the following.

$$\mathbb{E}_S \, \mathbb{E}_i \, e^{(2m-1)\gamma_i^2} \leq 4m. \tag{15}$$

By applying Markov's inequality on Eq. (15), we get the following.

$$P\left[ \sum \boldsymbol{S}_i e^{(2m-1)\gamma_i^2} \geq \frac{4m}{\delta} \right] \leq \delta, \tag{16}$$

which suffices to lemma 1.

To prove lemma 6 we consider selecting a training data distribution $\boldsymbol{S}$ and a predicted unseen-class data distribution $\boldsymbol{E}$. Lemma 1 implies that with probability at least $1 - \delta$ we have the following.

$$\sum \boldsymbol{S}_i e^{(2m-1)\gamma_i^2} \leq \frac{4m}{\delta}. \tag{17}$$

So to prove lemma 1 it now suffices to show that Eq. (17) plus $\ln \frac{1}{\delta} \leq 2m$ implies the following for all distributions $E$ such that $d(\boldsymbol{E}, \boldsymbol{S}) \leq 2m$.

$$\sum \boldsymbol{E}_i \gamma_i \leq \sum \boldsymbol{E}_i \sqrt{\frac{\ln \frac{\boldsymbol{E}_i}{\boldsymbol{S}_i} + \ln \frac{1}{\delta} + \frac{5}{2} \ln m + 8}{2m - 1}}. \tag{18}$$

To prove Eq. (18) for a given $\boldsymbol{E}$ we select $\gamma_i$ so as to maximize the quantity $\sum \boldsymbol{E}_i \gamma_i$ subject to the constraint Eq. (17). Using Lagrange multipliers we set the gradient of the constraint to be equal to a multiplier $\lambda$ times the gradient of the objective function.

$$2(2m-1)\gamma_i e^{(2m-1)\gamma_i^2} \boldsymbol{S}_i = \lambda \boldsymbol{E}_i. \tag{19}$$

Eq. (18) is trivially true if $E_i > 0$ but $S_i = 0$ for some $i$. So we can assume without loss of generality that $S_i > 0$ whenever $E_i > 0$. This allows the above to be rewritten as follows.

$$2(2m - 1)\gamma_i e^{(2m-1)\gamma_i^2} = \frac{\lambda E_i}{S_i}. \tag{20}$$

Note that $2(2m - 1)\gamma e^{(2m-1)\gamma^2}$ is an unbounded monotonically increasing function of $\gamma$. We now define $\triangle_i(\lambda)$ to be the unique non-negative value satisfying the following.

$$2(2m - 1)\triangle_i(\lambda) e^{(2m-1)\triangle_i^2(\lambda)} = \frac{\lambda E_i}{S_i}. \tag{21}$$

Now note that $\sum S_i e^{(2m-1)\triangle_i^2(\lambda)}$ is an unbounded monotonically increasing function of $\lambda$. We now define $\lambda^*$ to be the unique nonnegative value such that we have the following.

$$\sum S_i e^{(2m-1)\triangle_i^2(\lambda^*)} = \frac{4m}{\delta}. \tag{22}$$

Note that $\triangle_i(0) = 0$ and $\sum S_i e^{(2m-1)\triangle_i^2(0)} = 1 < \frac{4m}{\delta}$. So we must have $\lambda^* > 0$ and hence $\triangle_i(\lambda^*) > 0$ for $E_i > 0$. $\qquad\square$

**Lemma 2.** *For any $\gamma_i$ satisfying Eq. (17), we have the following.*

$$\sum E_i \gamma_i \leq E_i \triangle_i(\lambda^*). \tag{23}$$

*Proof.* Consider the following four situations:

(1) $\exists i \; \gamma_i < 0$

$\sum E_i \gamma_i$ can be increased by replacing $\gamma_i$ with $-\gamma_i$ for $\gamma_i < 0$. Hence we can assume without loss of generality that $\gamma_i \geq 0$.

(2) $\sum S_i e^{(2m-1)\gamma_i^2} < \frac{4m}{\delta}$

$\sum E_i \gamma_i$ can be increased by raising $\gamma_i$ with $E_i > 0$. Hence we can assume without loss of generality that $\sum S_i e^{(2m-1)\gamma_i^2} = \frac{4m}{\delta}$.

(3) $\exists i \; E_i = 0, \gamma_i > 0$

$\sum E_i \gamma_i$ can be increased by setting $\gamma_i = 0$ with $E_i = 0$ while raising $\gamma_i$ with $E_i > 0$. Hence we can assume without loss of generality that $\gamma_i = 0$ whenever $E_i = 0$.

(4) $\exists j, k \; E_j > 0, E_k > 0,$
$\frac{S_j}{E_j} \gamma_j e^{(2m-1)\gamma_j^2} > \frac{S_k}{E_k} \gamma_k e^{(2m-1)\gamma_k^2}$

Eq. (19) is trivially true if $E_i = 0$ and $\gamma_i = 0$. So we can rewrite Eq. (19) as follows.

$$\lambda_i = \frac{S_j}{E_i} 2(2m - 1)\gamma_i e^{(2m-1)\gamma_i^2}. \tag{24}$$

From Eq. (24) we can get the following.

$$\lambda_j = \frac{S_j}{E_j} 2(2m - 1)\gamma_j e^{(2m-1)\gamma_j^2} > \frac{S_k}{E_k} 2(2m - 1)\gamma_k e^{(2m-1)\gamma_k^2} = \lambda_k. \tag{25}$$

For $\lambda_i > 0$, Eq. (19) can be rewritten as follows.

$$E_i = \frac{S_j}{\lambda_i} 2(2m - 1)\gamma_i^2 e^{(2m-1)\gamma_i^2}. \tag{26}$$

So we have the following.

$$\sum E_i \gamma_i = \sum \frac{S_j}{\lambda_i} 2(2m - 1)\gamma_i^2 e^{(2m-1)\gamma_i^2}. \tag{27}$$

Let $f(x_i)$ be the function mapping $x_i$ to $\sum \frac{S_j}{\lambda_i} 2(2m-1)x_i e^{(2m-1)x_i}$.

$$f'(x_i) = 4m(2m-1)\sum \frac{S_i}{\lambda_i} x_i e^{(2m-1)x_i}. \tag{28}$$

For $x_j = x_k$, $f'(x_j) < f'(x_k)$.

$\sum E_i \gamma_i$ can be increased by increasing $\gamma_k$ and decreasing $\gamma_j$ while holding $\sum S_i e^{(2m-1)\gamma_i^2}$ constant. Hence we can assume without loss of generality that there exists a value $\lambda'$ such that for all indices $i$ with $E_i > 0$ we have $2(2m-1)\gamma_i e^{(2m-1)\gamma_i^2} = \frac{\lambda' E_i}{S_i}$, which implies $\gamma_i = \triangle_i(\lambda')$. We also have $\sum S_i e^{(2m-1)\triangle_i^2(\lambda')} = \frac{4m}{\delta}$, which implies $\lambda' = \lambda^*$. So we have $\gamma_i = \triangle_i(\lambda^*)$ which implies the result.

Now proving lemma 1 suffices to bound $\sum E_i \triangle_i(\lambda^*)$. Eq. (21) implies that for $\lambda \gg 1$ and $E_i \geq S_i$ we have the following

$$\triangle_i(\lambda) \approx \sqrt{\frac{\ln \frac{\lambda E_i}{S_i}}{2m-1}}. \tag{29}$$

$\square$

This approximate relationship is made more precise in the following two lemmas.

**Lemma 3.** *For $m \geq 1$, $S_i > 0$, $E_i \geq S_i$ and $\lambda \geq e$, we have the following.*

$$\triangle_i(\lambda) \leq \sqrt{\frac{\ln \frac{\lambda E_i}{S_i}}{2m-1}}. \tag{30}$$

*Proof.* Let $g(x)$ be the function mapping $x$ to $2(2m-1)xe^{(2m-1)x^2}$. By definition, $\triangle_i(\lambda)$ satisfies $g(\triangle_i(\lambda)) = \frac{\lambda E_i}{S_i}$. Note that for $x \geq 0$ we have that $g(x)$ is a monotonically increasing function. Hence for $x \geq 0$ and $g(x) \geq \frac{\lambda E_i}{S_i}$ we must have $\triangle_i(\lambda) \leq x$. Under the assumptions of lemma 3 we have $\ln \frac{\lambda E_i}{S_i} \geq 1$ which implies the following.

$$g(\sqrt{\frac{\ln \frac{\lambda E_i}{S_i}}{2m-1}}) = 2\sqrt{(2m-1)ln\frac{\lambda E_i}{S_i}}\frac{\lambda E_i}{S_i} \geq \frac{\lambda E_i}{S_i}. \tag{31}$$

$\square$

**Lemma 4.** *For $m \geq 1$, $S_i > 0$, $E_i \geq S_i$ and $\lambda \geq e$, we have the following.*

$$\triangle_i(\lambda) \geq \sqrt{\frac{\ln \frac{\lambda E_i}{S_i} + \frac{1}{2}\ln m - \ln\ln \frac{\lambda E_i}{S_i} - 2}{2m-1}}. \tag{32}$$

*Proof.* By an argument similar to that in the proof of lemma 3, to show that $\triangle_i(\lambda) \geq x$ it suffices to show that $g(x) \leq \frac{\lambda E_i}{S_i}$. In particular we have the following.

$$
\begin{aligned}
g(&\sqrt{\frac{\ln \frac{\lambda E_i}{S_i} + \frac{1}{2}\ln m - \ln\ln \frac{\lambda E_i}{S_i} - 2}{2m-1}}) \\
&= 2\sqrt{(2m-1)(\ln \frac{\lambda E_i}{S_i} + \frac{1}{2}\ln m - \ln\ln \frac{\lambda E_i}{S_i} - 2)} \\
&\quad \cdot \frac{\lambda E_i}{S_i}\frac{1}{\sqrt{m}\ln \frac{\lambda E_i}{S_i}e^2} \\
&\leq 4\sqrt{m\ln \frac{\lambda E_i}{S_i}}\frac{\lambda E_i}{S_i}\frac{1}{\sqrt{m}\ln \frac{\lambda E_i}{S_i}e^2} \\
&= \frac{\lambda E_i}{S_i}\frac{1}{\sqrt{\ln \frac{\lambda E_i}{S_i}}}\frac{4}{e^2} \\
&\leq \frac{\lambda E_i}{S_i}.
\end{aligned}
\tag{33}
$$

$\square$

**Lemma 5.** *For $d(\boldsymbol{E}, \boldsymbol{S}) \leq 2m$ and $\ln \frac{1}{\delta} \leq 2m$, we have $\lambda^* \leq \frac{64e^2 m^{5/2}}{\delta}$.*

*Proof.* Let $h(x)$ be the function mapping $x$ to $\sum \boldsymbol{S}_i e^{(2m-1)\triangle_i^2(x)}$. The quantity $\lambda^*$ is defined by $h(\lambda^*) = \frac{4m}{\delta}$. Since $h$ is a monotonically increasing function, $h(x) \geq \frac{2m}{\delta}$ implies $\lambda^* \leq x$. Lemma 4 implies that for $x \geq e$ we have the following.

$$
\begin{aligned}
h(x) &= \sum \boldsymbol{S}_i e^{(2m-1)\triangle_i^2(x)} \\
&\geq \sum \boldsymbol{S}_i \frac{x \boldsymbol{E}_i}{\boldsymbol{S}_i} \frac{1}{\sqrt{m} \ln \frac{x \boldsymbol{E}_i}{\boldsymbol{S}_i} e^2} \\
&= \frac{x}{e^2 \sqrt{m}} \sum \frac{\boldsymbol{E}_i}{\ln \frac{x \boldsymbol{E}_i}{\boldsymbol{S}_i}} \\
&\geq \frac{x}{e^2 \sqrt{m}} \frac{1}{\sum \boldsymbol{E}_i \ln \frac{x \boldsymbol{E}_i}{\boldsymbol{S}_i}} \\
&= \frac{x}{e^2 \sqrt{m}(d(E, B) + \ln x)} \\
&\geq \frac{x}{e^2 \sqrt{m}(2m + \ln x)}.
\end{aligned}
\tag{34}
$$

Now inserting $x = \frac{64e^2 m^{5/2}}{\delta}$ we get the following.

$$
\begin{aligned}
h(\frac{64e^2 m^{5/2}}{\delta}) &\geq \frac{64m^2}{\delta(2m + \frac{5}{2}\ln m + \ln \frac{1}{\delta} + 8)} \\
&\geq \frac{64m^2}{\delta(2m + \frac{5}{2}m + 2m + 8m)} \\
&\geq \frac{4m}{\delta}.
\end{aligned}
\tag{35}
$$

$\square$

**Lemma 6.** *Without loss of generality, we define the distance between distributions as $d(\boldsymbol{P}, \boldsymbol{Q}) = \sum \boldsymbol{P}_i \ln \frac{\boldsymbol{P}_i}{\boldsymbol{Q}_i}$ for analysis. For $\ln \frac{1}{\delta} \leq 2m$ we have that with probability $1 - \delta$ over the training dataset of size $m$ the following holds for all distributions satisfying $d(\boldsymbol{E}, \boldsymbol{S}) \leq 2m$.*

$$
d(\boldsymbol{E}, \boldsymbol{O}) - d(\boldsymbol{E}, \boldsymbol{S}) \leq \sum \boldsymbol{E}_i \sqrt{\frac{\ln \frac{\boldsymbol{E}_i}{\boldsymbol{S}_i} + \ln \frac{1}{\delta} + \frac{5}{2}\ln m + 8}{2m - 1}}.
\tag{36}
$$

*Proof.* Note that $d(\boldsymbol{E}, \boldsymbol{O}) - d(\boldsymbol{E}, \boldsymbol{S}) = \sum \boldsymbol{E}_i(\ln \frac{\boldsymbol{E}_i}{\boldsymbol{O}_i} - \ln \frac{\boldsymbol{E}_i}{\boldsymbol{S}_i}) \leq \sum \boldsymbol{E}_i \gamma_i$, where $\gamma_i$ abbreviates $|\ln \frac{\boldsymbol{E}_i}{\boldsymbol{O}_i} - \ln \frac{\boldsymbol{E}_i}{\boldsymbol{S}_i}|$, the lemma can be viewed as an upper bound of $\sum \boldsymbol{E}_i \gamma_i$.

From the above-mentioned lemmas, we have that

$$
\begin{aligned}
d(\boldsymbol{E}, \boldsymbol{O}) - d(\boldsymbol{E}, \boldsymbol{S}) &\leq \sum \boldsymbol{E}_i \gamma_i \\
&\leq \sum \boldsymbol{E}_i \triangle_i(\lambda^*) \\
&\leq \sum \boldsymbol{E}_i \triangle_i(\frac{64e^2 m^{5/2}}{\delta}) \\
&\leq \sum \boldsymbol{E}_i \sqrt{\frac{\ln \frac{64e^2 m^{5/2} \boldsymbol{E}_i}{\delta \boldsymbol{S}_i}}{2m - 1}} \\
&\leq \sum \boldsymbol{E}_i \sqrt{\frac{\ln \frac{\boldsymbol{E}_i}{\boldsymbol{S}_i} + \ln \frac{1}{\delta} + \frac{5}{2}\ln m + 8}{2m - 1}}.
\end{aligned}
\tag{37}
$$

$\square$

**Lemma 7.** *Denote the $d(\cdot, \cdot)$ as the distribution distance. With probability at least $1 - \delta$, we have the following,*

$$d\big(p(\boldsymbol{x}_o, \boldsymbol{y}_o), p(\boldsymbol{x}_e, \boldsymbol{y}_e)\big) \leq d\big(p(\boldsymbol{x}_e, \boldsymbol{y}_e), p(\boldsymbol{x}_s, \boldsymbol{y}_s)\big) + \sqrt{\frac{d\big(p(\boldsymbol{x}_e, \boldsymbol{y}_e), p(\boldsymbol{x}_s, \boldsymbol{y}_s)\big)}{2m - 1}}$$
$$+ \sqrt{\frac{\ln \frac{1}{\delta} + \frac{5}{2} \ln m + 8}{2m - 1}}, \tag{38}$$

*where $m$ denotes the size of training dataset.*

*Proof.* By applying Jensen's inequality on lemma 6, we can get Theorem 3.

$$d(\boldsymbol{E}, \boldsymbol{O}) \leq d(\boldsymbol{E}, \boldsymbol{S}) + \sum \boldsymbol{E}_i \sqrt{\frac{\ln \frac{\boldsymbol{E}_i}{\boldsymbol{S}_i} + \ln \frac{1}{\delta} + \frac{5}{2} \ln m + 8}{2m - 1}}$$
$$\leq d(\boldsymbol{E}, \boldsymbol{S}) + \sqrt{\frac{d(\boldsymbol{E}, \boldsymbol{S}) + \ln \frac{1}{\delta} + \frac{5}{2} \ln m + 8}{2m - 1}} \tag{39}$$
$$\leq d(\boldsymbol{E}, \boldsymbol{S}) + \sqrt{\frac{d(\boldsymbol{E}, \boldsymbol{S})}{2m - 1}} + \sqrt{\frac{\ln \frac{1}{\delta} + \frac{5}{2} \ln m + 8}{2m - 1}}.$$

$\square$

From Lemma 7, we can obviously observe that the distribution distance of the generated unseen-class data and the open-environment has an upper bound, which indicates that the rationality of generating unseen-class data for distribution estimation in open environments. Obviously, we also can observe that this upper bound is strongly related to the distribution distance between the generated unseen-class data and the seen-class data. The conclusion of Lemma 7 is same with Theorem 3. This also motivates us to construct the proposed open-vocabulary method. From Lemma 7, we can directly obtain Theorem 3.

**Theorem 3.** *Denote the $d(\cdot, \cdot)$ as the distribution distance. With probability at least $1 - \delta$, we have the following,*

$$d\big(p(\boldsymbol{x}_u, \boldsymbol{y}_u), p(\boldsymbol{x}_e, \boldsymbol{y}_e)\big) \leq \sqrt{\frac{d\big(p(\boldsymbol{x}_e, \boldsymbol{y}_e), p(\boldsymbol{x}_s, \boldsymbol{y}_s)\big)}{2m - 1}}$$
$$+ \sqrt{\frac{\ln \frac{1}{\delta} + \frac{5}{2} \ln m + 8}{2m - 1}}, \tag{40}$$

*where $m$ denotes the size of training dataset.*

## B   Proof of Theorem 4

In the manuscripts, we present Theorem 4 to demonstrate that the estimation error of $p(\bar{\boldsymbol{y}}|\boldsymbol{x}_u, \boldsymbol{\Phi})$ is upper bounded. Here we provide specific derivations of Theorem 4.

**Theorem 4.** *Given the predicted classes $\boldsymbol{Y}_e = \{\boldsymbol{y}_e\}$. Suppose that predicted class $\boldsymbol{Y}_e$ have any nonzero probability $p(\boldsymbol{Y}_e)$. With probability at least $1 - \delta$ over the $m$ instances of generated unseen-class data $\{(\boldsymbol{x}_e, \boldsymbol{y}_e)\}$, we have that*

$$D_{KL}\big(p(\bar{\boldsymbol{y}}|\boldsymbol{x}_e, \boldsymbol{\Phi})||p(\bar{\boldsymbol{y}}|\boldsymbol{x}_u, \boldsymbol{\Phi})\big) \leq \sqrt{\frac{\ln \frac{1}{p(\boldsymbol{Y}_e)} + \ln \frac{1}{\delta}}{2m}} - \ln \frac{|\boldsymbol{Y}_e|}{|\boldsymbol{Y}_u|}. \tag{41}$$

*Proof.* We denote $p^*(\boldsymbol{y}) = p(\bar{\boldsymbol{y}}|\boldsymbol{x}_u, \boldsymbol{\Phi}), \bar{\boldsymbol{y}} \in \boldsymbol{Y}_u$. For analysis, we define $P(\boldsymbol{Y}_e) = \sum\limits_{\boldsymbol{y} \in \boldsymbol{Y}_e} p^*(\boldsymbol{y})$, and

we define the conditional distribution as $P_e(\boldsymbol{y}) = p(\bar{\boldsymbol{y}}|\boldsymbol{x}_u, \boldsymbol{\Phi}) = \frac{p^*(\boldsymbol{y})}{P(\boldsymbol{Y}_e)}, \bar{\boldsymbol{y}} \in \boldsymbol{Y}_e$. $P_e(\boldsymbol{y})$ satisfies

that $\sum_{\boldsymbol{y}\in\boldsymbol{Y}_e} P_e(\boldsymbol{y}) = 1$. We also denote that $\hat{p}(\boldsymbol{y}) = p(\bar{\boldsymbol{y}}|\boldsymbol{x}_e, \boldsymbol{\Phi})$. In this way, the KL divergence between $\hat{p}(\boldsymbol{y})$ and $p^*(\boldsymbol{y})$ is computed as

$$D_{\text{KL}}(\hat{p}(\boldsymbol{y})||p^*(\boldsymbol{y})) = \sum_{\boldsymbol{y}\in\boldsymbol{Y}_e} \hat{p}(\boldsymbol{y})\ln\frac{\hat{p}(\boldsymbol{y})}{p^*(\boldsymbol{y})}. \tag{42}$$

$\frac{\hat{p}(\boldsymbol{y})}{p^*(\boldsymbol{y})}$ can be formulated as

$$\frac{\hat{p}(\boldsymbol{y})}{p^*(\boldsymbol{y})} = \frac{\hat{p}(\boldsymbol{y})}{P_e(\boldsymbol{y})}\frac{P_e(\boldsymbol{y})}{p^*(\boldsymbol{y})} = \frac{\hat{p}(\boldsymbol{y})}{P_e(\boldsymbol{y})}\frac{1}{P(\boldsymbol{Y}_e)}, \tag{43}$$

and thus

$$\ln\frac{\hat{p}(\boldsymbol{y})}{p^*(\boldsymbol{y})} = \ln\frac{\hat{p}(\boldsymbol{y})}{P_e(\boldsymbol{y})} - \ln P(\boldsymbol{Y}_e). \tag{44}$$

By substituting Eq. (44) into Eq (42), we have that

$$\begin{aligned} D_{\text{KL}}(\hat{p}(\boldsymbol{y})||p^*(\boldsymbol{y})) &= \sum_{\boldsymbol{y}\in\boldsymbol{Y}_e} \hat{p}(\boldsymbol{y})\left(\ln\frac{\hat{p}(\boldsymbol{y})}{P_e(\boldsymbol{y})} - \ln P(\boldsymbol{Y}_e)\right) \\ &= \sum_{\boldsymbol{y}\in\boldsymbol{Y}_e} \hat{p}(\boldsymbol{y})\ln\frac{\hat{p}(\boldsymbol{y})}{P_e(\boldsymbol{y})} - \sum_{\boldsymbol{y}\in\boldsymbol{Y}_e} \hat{p}(\boldsymbol{y})\ln P(\boldsymbol{Y}_e) \\ &= D_{\text{KL}}(\hat{p}(\boldsymbol{y})||P_e(\boldsymbol{y})) - \ln P(\boldsymbol{Y}_e). \end{aligned} \tag{45}$$

By the Chernoff bound, for the generated unseen-class data $\{(\boldsymbol{x}_e, \boldsymbol{y}_e)\}$, we have that

$$P\left[D_{\text{KL}}(\hat{p}(\boldsymbol{y})||P_e(\boldsymbol{y})) \geq \delta\right] \leq e^{-2mt^2}. \tag{46}$$

From the union bound, and the classes are countable, we have that

$$P[\exists\boldsymbol{Y}_e : D_{\text{KL}}(\hat{p}(\boldsymbol{y})||P_e(\boldsymbol{y})) \geq t] \leq \sum_{\boldsymbol{Y}_e} P[D_{\text{KL}}(\hat{p}(\boldsymbol{y})||P_e(\boldsymbol{y})) \geq t]. \tag{47}$$

Assign the probability $p(\boldsymbol{Y}_e)$ for $\boldsymbol{Y}_e$, and thus $\sum_{\boldsymbol{Y}_e} p(\boldsymbol{Y}_e) = 1$. In this way, we have that

$$P[\exists\boldsymbol{Y}_e : D_{\text{KL}}(\hat{p}(\boldsymbol{y})||P_e(\boldsymbol{y})) \geq \delta] \leq \sum_{\boldsymbol{Y}_e} e^{-2mt^2} = \sum_{\boldsymbol{Y}_e} p(\boldsymbol{Y}_e)\frac{e^{-2mt^2}}{p(\boldsymbol{Y}_e)}. \tag{48}$$

Letting $e^{-2mt^2} = p(\boldsymbol{Y}_e)\delta$, we have that

$$P[\exists\boldsymbol{Y}_e : D_{\text{KL}}(\hat{p}(\boldsymbol{y})||P_e(\boldsymbol{y})) \geq t] \leq \sum_{\boldsymbol{Y}_e} p(\boldsymbol{Y}_e)\delta = \delta, \tag{49}$$

where $t$ is computed as

$$t = \sqrt{\frac{\ln\frac{1}{p(\boldsymbol{y}_e^i)} + \ln\frac{1}{\delta}}{2m}}. \tag{50}$$

In this way, we can derive that

$$P\left[D_{\text{KL}}(\hat{p}(\boldsymbol{y})||P_e(\boldsymbol{y})) \geq \sqrt{\frac{\ln\frac{1}{p(\boldsymbol{y}_e^i)} + \ln\frac{1}{\delta}}{2m}}\right] \leq \delta, \tag{51}$$

Therefore, with probability at least $1 - \delta$, we have that

$$D_{\text{KL}}(\hat{p}(\boldsymbol{y})||P_e(\boldsymbol{y})) \leq \sqrt{\frac{\ln\frac{1}{p(\boldsymbol{y}_e^i)} + \ln\frac{1}{\delta}}{2m}}. \tag{52}$$

We substitute Eq. (52) into Eq. (45). With probability at least $1 - \delta$, we have that

$$D_{\text{KL}}\left(p(\bar{\boldsymbol{y}}|\boldsymbol{x}_e, \boldsymbol{\Phi})||p(\bar{\boldsymbol{y}}|\boldsymbol{x}_u, \boldsymbol{\Phi})\right) \leq \sqrt{\frac{\ln\frac{1}{p(\boldsymbol{Y}_e)} + \ln\frac{1}{\delta}}{2m}} - \ln\frac{|\boldsymbol{Y}_e|}{|\boldsymbol{Y}_u|}. \tag{53}$$

$\square$

## C  Proof of ELBO in Eq. (7) in the Manuscripts

**Proposition 1.** *The Evidence Lower Bound (ELBO) of the logarithmic posterior probability can be derived as*

$$\log p(\bar{\boldsymbol{y}}|\boldsymbol{x}_o, \boldsymbol{\Phi}) \geq \mathbb{E}[\log p(\bar{\boldsymbol{y}}|\boldsymbol{x}_s, \boldsymbol{\Phi})] - D_{KL}(p(\bar{\boldsymbol{y}}|\boldsymbol{x}_s, \boldsymbol{\Phi})||p(\bar{\boldsymbol{y}}|\boldsymbol{x}_u, \boldsymbol{\Phi})), \tag{54}$$

*Proof.* The probability $p(\bar{\boldsymbol{y}}|\boldsymbol{x}_o)$ can be modeled as a function $f(\boldsymbol{y})$. Therefore, the logarithmic probability $\log p(\boldsymbol{y}|\boldsymbol{x}_o)$ is computed as

$$\log p(\boldsymbol{y}|\boldsymbol{x}_o) = \log f(\boldsymbol{y}) = \log \left( \int f(\tilde{\boldsymbol{y}})\delta(\tilde{\boldsymbol{y}} - \boldsymbol{y})d\tilde{\boldsymbol{y}} \right), \tag{55}$$

where $\delta(\cdot)$ is a Dirac function, and $\tilde{\boldsymbol{y}}$ is a intermediate variable. The last equality holds since the proposition of the Dirac function. We introduce a variational distribution $q(\tilde{\boldsymbol{y}})$. Then, $\log f(\boldsymbol{y})$ holds that

$$\log f(\boldsymbol{y}) = \log \left( \int f(\tilde{\boldsymbol{y}})\delta(\tilde{\boldsymbol{y}} - \boldsymbol{y})d\tilde{\boldsymbol{y}} \right) = \log \left( \int f(\tilde{\boldsymbol{y}})\frac{q(\tilde{\boldsymbol{y}})}{q(\tilde{\boldsymbol{y}})}\delta(\tilde{\boldsymbol{y}} - \boldsymbol{y})d\tilde{\boldsymbol{y}} \right)$$
$$= \log \left( \int q(\tilde{\boldsymbol{y}})\frac{f(\tilde{\boldsymbol{y}})\delta(\tilde{\boldsymbol{y}} - \boldsymbol{y})}{q(\tilde{\boldsymbol{y}})}d\tilde{\boldsymbol{y}} \right). \tag{56}$$

From the Jensen inequality, we can derive that

$$\log f(\boldsymbol{y}) \geq \int q(\tilde{\boldsymbol{y}}) \log \frac{f(\tilde{\boldsymbol{y}})\delta(\tilde{\boldsymbol{y}} - \boldsymbol{y})}{q(\tilde{\boldsymbol{y}})}d\tilde{\boldsymbol{y}} = \mathbb{E}_{q(\boldsymbol{y})}\left[ \log \frac{f(\boldsymbol{y})\delta(\tilde{\boldsymbol{y}} - \boldsymbol{y})}{q(\boldsymbol{y})} \right]. \tag{57}$$

To guarantee the boundedness of the ELBO, we ignore the Dirac function. Eq. (57) can be further modeled as

$$\log f(\boldsymbol{y}) \geq \mathbb{E}_{q(\boldsymbol{y})}\left[ \log f(\boldsymbol{y}) \right] - \mathbb{E}_{q(\boldsymbol{y})}\left[ \log q(\boldsymbol{y}) \right]. \tag{58}$$

Then, substituting $f(\boldsymbol{y}) = \log p(\bar{\boldsymbol{y}}|\boldsymbol{x}_o, \boldsymbol{\Phi})$ and $q(\boldsymbol{y}) = p(\bar{\boldsymbol{y}}|\boldsymbol{x}_s, \boldsymbol{\Phi})$ , we have that

$$\log p(\bar{\boldsymbol{y}}|\boldsymbol{x}_o, \boldsymbol{\Phi}) \geq \mathbb{E}_{p(\bar{\boldsymbol{y}}|\boldsymbol{x}_s, \boldsymbol{\Phi})}\left[ \log p(\bar{\boldsymbol{y}}|\boldsymbol{x}_s, \boldsymbol{\Phi}) \right] + \mathbb{E}_{p(\bar{\boldsymbol{y}}|\boldsymbol{x}_s, \boldsymbol{\Phi})}\left[ \log p(\bar{\boldsymbol{y}}|\boldsymbol{x}_u, \boldsymbol{\Phi}) \right]$$
$$- \mathbb{E}_{p(\bar{\boldsymbol{y}}|\boldsymbol{x}_s, \boldsymbol{\Phi})}\left[ \log p(\bar{\boldsymbol{y}}|\boldsymbol{\Phi}) \right] - \mathbb{E}_{p(\bar{\boldsymbol{y}}|\boldsymbol{x}_s, \boldsymbol{\Phi})}\left[ \log p(\bar{\boldsymbol{y}}|\boldsymbol{x}_s, \boldsymbol{\Phi}) \right]$$
$$= \mathbb{E}_{p(\bar{\boldsymbol{y}}|\boldsymbol{x}_s, \boldsymbol{\Phi})}\left[ \log p(\bar{\boldsymbol{y}}|\boldsymbol{x}_s, \boldsymbol{\Phi}) \right] + \mathbb{E}_{p(\bar{\boldsymbol{y}}|\boldsymbol{x}_s, \boldsymbol{\Phi})}\left[ \log \frac{p(\bar{\boldsymbol{y}}|\boldsymbol{x}_u, \boldsymbol{\Phi})}{p(\bar{\boldsymbol{y}}|\boldsymbol{x}_s, \boldsymbol{\Phi})} \right] \tag{59}$$
$$- \mathbb{E}_{p(\bar{\boldsymbol{y}}|\boldsymbol{x}_s, \boldsymbol{\Phi})}\left[ \log p(\bar{\boldsymbol{y}}|\boldsymbol{\Phi}) \right].$$

The probability term $p(\bar{\boldsymbol{y}}|\boldsymbol{\Phi})$ can be ignored because it appears as a constant term in the variational lower bound (ELBO). It depends only on the model parameters $\boldsymbol{\Phi}$ and is independent of the input data $\boldsymbol{x}$. As such, it does not affect the gradient computation or the update of model parameters $\boldsymbol{\Phi}$. Since this term does not contribute to the optimization process, it can be safely omitted, simplifying the derivation. In many works, derivation of ELBO commonly omit such constant terms, as they do not affect the optimization objective and can be safely ignored to simplify the computation [**?, ?, ?**]. Therefore, we model Eq. (59) as

$$\log p(\bar{\boldsymbol{y}}|\boldsymbol{x}_o, \boldsymbol{\Phi}) \geq \mathbb{E}\left[ \log p(\bar{\boldsymbol{y}}|\boldsymbol{x}_s, \boldsymbol{\Phi}) \right] - D_{\text{KL}}(p(\bar{\boldsymbol{y}}|\boldsymbol{x}_s, \boldsymbol{\Phi})||p(\bar{\boldsymbol{y}}|\boldsymbol{x}_u, \boldsymbol{\Phi})). \tag{60}$$

$\square$

## D  Details and Illustration of Class-Domain-Wise Data Generation Pipeline

### D.1  Specific Details of Hierarchy-Guided Unseen Class Predictor

The hierarchy-guided unseen class predictor identifies the potential unseen classes, which are close to training classes. This is achieved by constructing a hierarchical semantic tree, where leaf nodes represent training classes and parent nodes represent their superclasses. The tree is expanded by adding leaf nodes of the candidate unseen classes sourced from WordNet or LLMs. As illustrated in Figure 1, given the training classes "Goldfish," "Tench," and "Ray," a hierarchical semantic tree is constructed where these classes are set as leaf nodes and their superclass "Fish" is set as a parent node. After LLMs are queried, "Salmon" is added as a leaf node of the candidate unseen class under

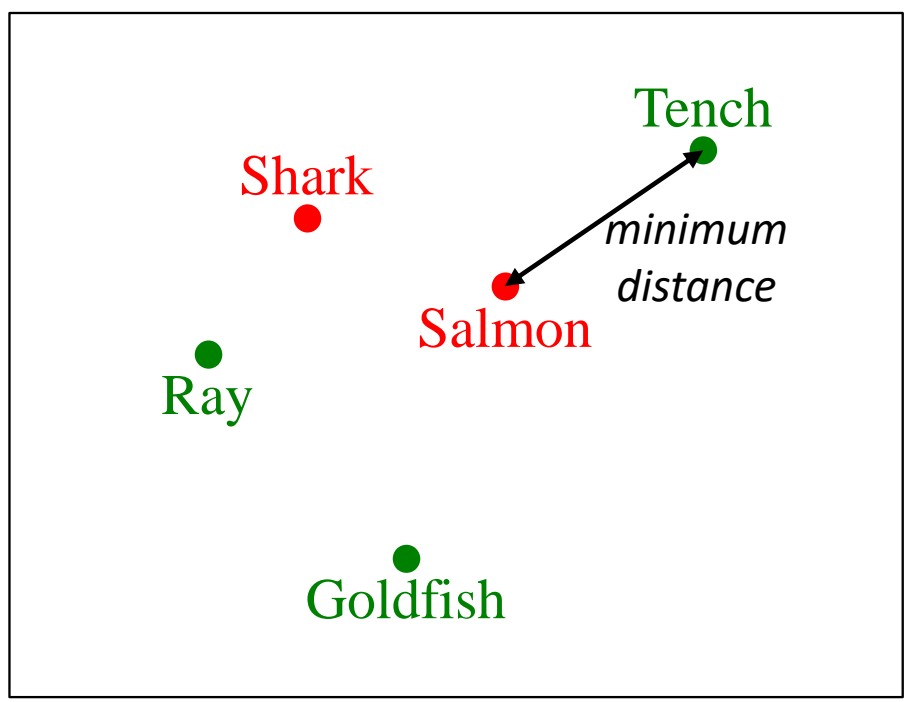

Figure 3: Distance between Classes

the "Fish" superclass. To select the closest candidate unseen class, the predictor computes the cosine similarity between the textual embeddings of candidate unseen classes and given training classes from the text encoder of a pretrained CLIP. The top $K$ closest candidates are chosen as the identified potential unseen classes. Take the classes shown in Figure 3 as an example, the cosine similarity between the textual embeddings of "Tench" and candidate unseen classes "Salmon", "Shark" is computed. The candidate unseen class with the highest similarity "Salmon" is chosen as a predicted unseen class.

### D.2 Specific Details of Caption-Based Domain Information Generator

The caption-based domain information generator extracts contextual attributes, such as styles and backgrounds, from the training data to ensure the generated unseen data align with the visual characteristics of training data. This is achieved by generating class-specific captions for each training class using VLMs. The generator then computes the similarity between these captions and the corresponding data, selecting the top $K_1$ captions with the highest similarity to mitigate hallucination issues. These selected captions are further summarized into top $K_2$ class-specific domain information using LLMs. Finally, the predicted unseen classes and the summarized domain information are combined into textual prompts, which guide the data generation process using a text-to-image model such as Stable Diffusion. In Figure 1, for instance, captions for training images of "Tench" are first generated using VLMs. The generator then calculates the similarity between these captions and the corresponding images of "Tench". The top 3 captions with the highest similarity are selected, which describe the domain information such as "holded by man", "golden-colored" and "prominent eye". These captions are summarized into top 1 class-specific domain information using LLMs, which is presented in the caption template in the red box. Finally, the predicted unseen class "Salmon" is inserted into the template to create an image caption, which is then used as input for Stable Diffusion to generate images of "Salmon".

### D.3 Prompt Templates

In the manuscripts, we propose to utilize LLMs and VLMs to identify potential unseen classes and extract domain information of training data. Here we provide the utilized 3 prompt templates.

As to the unseen class predictor, we aim to query the Hypernym of training classes by the following prompt. We first construct the in-context examples for accurate results, as shown in Template 1.

**Template 1.** *{class} denotes the training class.*
*Q: What is the Hypernym category of {class1}?*
*A: {class2} is the Hypernym category of {class1}.*

The prompt is given in Template 2.

**Template 2.** *{class} denotes the training class.*
*Q: What is the Hypernym category of {class3}?*

Then, with the generated Hypernym, we leverage LLMs to identify potential unseen classes using LLMs, where the in-context examples and prompts are shown in template 3 and template 4, respectively.

**Template 3.** *{class} denotes the training class.*
*Q: What is the Hyponym category of {class1}?*
*A: {class2} is the Hyponym category of {class1}.*

**Template 4.** *{class} denotes the training class.*
*Q: What is the Hyponym category of {class3}?*

We leverage template 5 to generate class-specific captions for each training class using VLMs.

**Template 5.** *{class} denotes the training class.*
*user prompt:*
*This is an image of {class}. Summarize the main style, scene, and key elements of this image in one sentence.*

We leverage template 6 to summarize captions into class-specific domain information using LLMs

**Template 6.** *{class} denotes the predicted unseen class.*
*system prompt:*
*As a caption summarizer, your task is to transform the provided captions from their original category to a new specified category and condense them into a concise set of 3 distinct one-sentence captions. Make sure the new captions maintain coherence with the original style but reflect the characteristics of the new target category. Each caption must capture a unique artistic style or visual theme. Only generate the transformed one-sentence captions—no introductions, explanations, or comments. The output should strictly follow this format:*
*1. [Caption 1]*
*2. [Caption 2]*
*3. [Caption 3]*
*user prompt:*
*Transform and condense the following captions into 3 new one-sentence captions describing {class}, each focusing on a distinct artistic style or visual theme.*

## E   Distribution Alignment

In this paper, we adopt prompt learning method to optimize the pretrained model. We propose a distribution alignment algorithm which aligns the output distributions of model on seen-class data and generated unseen-class data to maximize the logarithmic posterior probability in open environments. The proposed algorithm is summarized in Algorithm 1.

## F   Experiment Details in Manuscripts

In this section, we provide more specific details of experiments in the manuscripts. Specifically, we present the specific implementation details, details and extra analysis in ablation studies of the manuscripts.

**Algorithm 1** Distribution Alignment Algorithm

---

**Input**: Parameters $\Phi$, Data $\mathcal{D}_s, \mathcal{G}_u, \mathcal{G}_s$, Epoch $E$, batch-size B
**Output**: Optimized Prompts $\boldsymbol{v}_{max\_iter}$
**Initialize**: e = 0, $\boldsymbol{v} \leftarrow \boldsymbol{v}_0$, $S \leftarrow \{\}$

1: **while** $e \leq E$ **do**
2:     **for** $i = 1, 2, ..., |\mathcal{D}_s|/B$ **do**
3:         Compute the posterior probability of model output in the current batch of the seen-class dataset $\mathcal{D}_s^i$.
4:         Accumulate the output posterior probability for distribution alignment into $S$.
5:         **if** $i \% 8 == 0$ **then**
6:            $S_{KL} = \{\}$.
7:            **for** $j = 1, 2, ..., |\mathcal{G}_u|/B$ **do**
8:                Compute the KL divergence between the accumulated posterior probability on the seen-class data and the mini-batch of generated unseen-class data $d_{kl} = D_{KL}[p(\bar{\boldsymbol{y}}|\boldsymbol{x}_s, \boldsymbol{\Phi}, \boldsymbol{v})||p(\bar{\boldsymbol{y}}|\boldsymbol{x}_e, \boldsymbol{\Phi}, \boldsymbol{v})]$.
9:                Update the set as $S_{KL}.append(d_{kl})$.
10:            **end for**
11:            Compute $\mathcal{L}_{KL}$ based on top $K_3$ smallest in the set $S_{KL}$ as $\mathcal{L}_{KL} = \frac{1}{K_3} \sum_{topK_3} d_{kl}$.
12:            $S_{mmd} = \{\}$.
13:            **for** $m = 1, 2, ..., |\mathcal{G}_s|/B$ **do**
14:                Compute MMD loss $l_{mmd}$ based on the generated unseen-class data and the seen-class data on the current batch, and save them into $S_{mmd}$.
15:            **end for**
16:            Compute $\mathcal{L}_{MMD}$ based on top $K_3$ smallest $l_{mmd}$ as $\mathcal{L}_{MMD} = \frac{1}{K_3} \sum_{topK_3} l_{mmd}$.
17:            Compute total loss $\mathcal{L}_{total} = \mathcal{L}_{CE} + \alpha \mathcal{L}_{KL} + \beta \mathcal{L}_{MMD}$.
18:            Backward and update the prompt $\boldsymbol{v}$ using $\mathcal{L}_{CE}$ and $\mathcal{L}_{total}$.
19:            Clear saved data $S = \{\}$.
20:         **else**
21:            Compute $\mathcal{L}_{CE}$ on the mini-batch of seen-class dataset $\mathcal{D}_s^i$.
22:            Backward and update the prompt $\boldsymbol{v}$ using $\mathcal{L}_{CE}$.
23:         **end if**
24:     **end for**
25: **end while**
26: **return** The updated prompts $\boldsymbol{v}$.

---

### F.1 Implementation details

For base-to-base/base-to-new generalization, we train each model for 20 epochs using 4 token prompts in the first 9 transformer layers on both visual and text branch. For cross-dataset evaluation, we train the source model for 4 epochs using 4 prompts in the first 3 transformer layers on both visual and text branch. Prompts are randomly initialized with a normal distribution except the text prompts of the first layer which are initialized with the word embeddings of "a photo of a". The SGD optimizer is adopted, and the learning rate is set as 0.0025. Hyperparameters for the class-domain-wise data generation pipeline and distribution alignment are determined empirically. Specifically, We set $\alpha = 1$, $\beta = 1$, $K_0$ as 1, $K_1$ as 8, $K_2$ as 3 and $K_3$ as 1. The corresponding hyperparameters are fixed across all datasets and benchmarks.

For LLMs and VLMs, we use Doubao-pro-128k to identifies the potential unseen classes, use LLaVA-v1.6-Vicuna-13B [39] to generate class-specific captions for each training class, use Llama-v3.1-Instruct-8B [61] to summarize captions into class-specific domain information, and use Stable Diffusion v2.1 [50] as the text-to-image model to generate unseen-class data.

Experiments are performed on an NVIDIA A40 GPU, with at most 18 hours 20 GPU memory required to complete training across 11 datasets.

Table 5: Ablation study on sparse loss computation strategy. "w/o spa" denotes distribution alignment algorithm without sparse loss computation strategy.

| | Caltech | | Pets | | Cars | | Flowers | | Food | | Aircraft | | DTD | | EuroSAT | | UCF | |
|---|---|---|---|---|---|---|---|---|---|---|---|---|---|---|---|---|---|---|
| | w/o spa | Ours | w/o spa | Ours | w/o spa | Ours | w/o spa | Ours | w/o spa | Ours | w/o spa | Ours | w/o spa | Ours | w/o spa | Ours | w/o spa | Ours |
| Base | 98.52 | **98.97** | 93.36 | **96.01** | 81.88 | **82.93** | 96.68 | **98.77** | 91.14 | **91.39** | 46.40 | **48.98** | 82.41 | **85.53** | 95.95 | **97.17** | 87.07 | **89.14** |
| New | 94.21 | **95.85** | 91.33 | **97.65** | 79.48 | **80.81** | 78.09 | **80.92** | 92.56 | **92.99** | 39.71 | **44.03** | 61.72 | **71.50** | 76.92 | **87.90** | 79.61 | **82.53** |
| H | 96.32 | **97.38** | 92.33 | **96.82** | 80.66 | **81.86** | 86.39 | **88.96** | 91.84 | **92.18** | 42.80 | **46.37** | 70.58 | **77.89** | 85.39 | **92.30** | 83.17 | **85.71** |

## F.2 Details and Extra Analysis of Ablation Study

In this subsection, we present the details in ablation studies in hierarchy-guided unseen class predictor and caption-based domain information generator. Then, we present the extra analysis in ablation studies of the manuscripts.

### F.2.1 Details of Hierarchy-Guided Unseen Class Predictor

To evaluate the effectiveness of hierarchy-guided unseen class predictor, we first investigate how the quantity of predicted unseen classes impacts the model performance in open environments. Specifically, we introduce a class sampling ratio $s_{cls}$ to control the quantity of predicted unseen classes used for training. Assume that we initially generate $N$ unseen classes for a given dataset, with each class containing $M$ images, we randomly select $s_{cls} \times N$ classes from the predicted $N$ unseen classes and then train the model using $s_{cls} \times N \times M$ images of these classes, discarding the remaining classes and their images.

Next, we investigate how the quality of predicted unseen classes impacts the model performance in open environments. "Low Similarity" denotes that when predicting unseen classes, we compute the cosine similarity to textual seen classes for each candidate class and choose the one with the lowest cosine similarity as the predicted unseen class. "w/oTree" denotes that instead of constructing a hierarchical semantic tree to predict unseen classes, we directly ask LLMs to provide a predicted unseen class corresponding to the given base classes.

### F.2.2 Details of Caption-Based Domain Information Generator

To evaluate the effectiveness of caption-based domain information generator, we first investigate how the quantity of generated unseen-class images impacts the model performance in open environments. Similarly, we introduce a image sampling ratio $s_{img}$ to control the number of generated images of predicted unseen classes used for training. For a given dataset, we initially predict $N$ unseen classes with each class containing $M$ generated images. Based on the image sampling ratio $s_{img}$, we randomly select $s_{img} \times M$ images from each predicted unseen class. The selected $N \times s_{img} \times M$ images from $N$ unseen classes are then used for training, while the remaining images are discarded.

Next, we investigate how the quality of generated images from predicted unseen classes impacts the model performance in open environments. We modify the prompts used in our unseen image generator to control the quality of image generation. In this work, we use the prompts "A picture of a category ", "A photo of a category ", and "An image of a category " as templates for the Stable Diffusion model when generating images of unseen classes, respectively.

## F.3 Extra Analysis

The ablation studies provide a comprehensive analysis of the relationship between distribution distance and accuracy, which aligns with the theoretical analysis. Recall that Theorem 3 shows that the estimation error between the unseen-class data distribution and the generated unseen-class data distribution is upper-bounded, and reducing the distribution gap between generated unseen-class data and seen-class data tightens this bound, thereby improving model performance in open environments. Experimental results from the ablation studies validate this theoretical claim. Specifically, as the distribution distance decreases, accuracy consistently improves across various datasets, particularly for new classes. This confirms that reducing the distribution gap between generated unseen-class and seen-class data leads to more accurate estimation of unseen-class data distribution, enhancing the model's generalization ability in open-vocabulary learning tasks.

# G  Robust Analysis

## G.1  Hyperparameter Analysis

We add experiments to analyze the sensitivity of hyperparameters $K_0, K_1, K_2, K_3$ in the data generation pipeline and $\alpha, \beta$ in the loss function. Specifically, we conduct experiments on the EuroSAT dataset by adjusting these hyperparameters. Results on $K_0, K_1, K_2, K_3$ are shown in Table 6 7 8 9, respectively. Results of $\alpha, \beta$ are presented in Table 10. Results show that the performance remains relatively stable with varying hyperparameter values, indicating that the method is only minimally sensitive to hyperparameter variation.

Table 6: $K_0$ Analysis

|      | $K_0 = 1$ | $K_0 = 5$ | $K_0 = 10$ | $K_0 = 20$ |
|------|-----------|-----------|------------|------------|
| Base | 97.17     | 96.64     | 96.43      | 96.24      |
| New  | 87.90     | 87.07     | 85.85      | 85.84      |
| H    | 92.30     | 91.61     | 90.83      | 90.57      |

Table 7: $K_1$ Analysis

|      | $K_1 = 8$ | $K_1 = 6$ | $K_1 = 4$ | $K_1 = 2$ |
|------|-----------|-----------|-----------|-----------|
| Base | 97.17     | 97.02     | 96.95     | 96.83     |
| New  | 87.90     | 86.31     | 86.0      | 85.85     |
| H    | 92.30     | 91.35     | 91.15     | 91.01     |

Table 8: $K_2$ Analysis

|      | $K_2 = 3$ | $K_2 = 2$ | $K_2 = 1$ |
|------|-----------|-----------|-----------|
| Base | 97.17     | 96.92     | 96.79     |
| New  | 87.90     | 86.59     | 85.31     |
| H    | 92.30     | 91.47     | 90.69     |

Table 9: $K_3$ Analysis

|      | $K_2 = 1$ | $K_2 = 2$ | $K_2 = 3$ |
|------|-----------|-----------|-----------|
| Base | 97.17     | 96.33     | 96.60     |
| New  | 87.90     | 85.36     | 86.31     |
| H    | 92.30     | 90.51     | 91.16     |

Table 10: $\alpha/\beta$ analysis ($\alpha + \beta = 1$)

| $\alpha$ | $\beta$ | Base  | New   | H     |
|----------|---------|-------|-------|-------|
| 0.3      | 0.7     | 96.88 | 86.15 | 91.20 |
| 0.4      | 0.6     | 96.86 | 86.26 | 91.25 |
| 0.5      | 0.5     | 97.17 | 87.90 | 92.30 |
| 0.6      | 0.4     | 96.81 | 87.06 | 91.68 |

## G.2  Robustness to Long-tailed Setting and Noise

**Robustness to long-tailed setting.** We construct a long-tail distribution setting by removing a portion of samples from selected classes and conduct comparative experiments (with the number of samples per class being 16, 16, 16, 10, and 2, respectively). In this setting, we evaluate the performance of our current top alignment strategy against a sample duplication strategy designed for long-tail settings. The results in Table 11 demonstrate that in the long-tailed setting, the sample duplication strategy can improve performance.

Table 11: Experiments on long-tailed benchmark

| | Our method (with top $K_g$ alignment) | Our method (with top $K_g$ alignment + sample duplication) |
|---|---|---|
| Base | 95.79 | 97.00 |
| New | 85.39 | 87.39 |
| H | 90.29 | 91.94 |

Table 12: Robustness evaluation to the noisy semantic tree

| | Correct Superclasses | Wrong Superclasses | PromptSRC (Baseline) |
|---|---|---|---|
| Base | 97.17 | 96.17 | 92.90 |
| New | 87.90 | 86.49 | 73.90 |
| H | 92.30 | 91.07 | 82.32 |

**Robustness to noise.** To verify the robustness, we conduct an experiment where we select incorrect superclasses on the EuroSAT dataset. Results in Table 12 show that the performance slightly degrades, demonstrating the robustness of our method.

# H   Extra Ablation Studies

In this section, we present extra ablation studies for validating the effectiveness of sparse loss computation strategy in distribution alignment algorithm. We demonstrate the effectiveness of the sparse loss computation strategy by conducting experiments on the distribution alignment algorithm without it, denoted as 'w/o spa'. The results, shown in Table 13, reveal that the sparse loss computation strategy significantly improves performance, particularly on the new classes. Notably, on the Pets, Cars, DTD, and EuroSAT datasets, the strategy achieves improvements of 6.32%, 9.18%, 9.78%, and 10.98% on the new classes compared to 'w/o spa'. These results further confirm the effectiveness of the proposed strategy.

# I   Visualization

In this section, we visualize the unseen-class images generated to demonstrate the effectiveness of the proposed class-domain-wise data generation pipeline. We compare the proposed method with three prompt templates for the text-to-image model mentioned in the ablation studies, namely, 'A picture of a class', 'A photo of class', and 'An image of a class'. We use the images generated based on the Caltech101 dataset for analysis.

We use the caption-based domain information generator to capture the class-specific domain information of seen-class data. This domain information is then used to generate the corresponding seen-class data via a text-to-image model. For visualization, we adopt the seen classes 'motorbike' and 'barrel'. As shown in Figures 4 and 5, compared to the three commonly used prompt templates, the generated seen-class data from the proposed pipeline better align with the seen-class data in terms of both style and scene information. This demonstrates that our pipeline is more effective at capturing the domain information of seen-class images for data generation.

Regarding the generation of unseen-class data, we use the hierarchy-based unseen class predictor to infer that the unseen classes 'car' and 'drum' are closest to 'motorbike' and 'barrel', respectively. The captured class-specific domain information and inferred unseen classes are then used to generate the unseen-class images via a text-to-image model. We compare the generated unseen-class data from our pipeline with data generated using the three commonly used prompt templates. As shown in Figures 4 and 5, the generated unseen-class data align better with the seen-class data. For example, in Figure 4, the car generated by our pipeline reflects the style of the seen-class data, and the realistic scene depicted in the generated images mirrors the scene in the seen-class data. These results further demonstrate that our pipeline effectively captures the domain information of seen-class images, and the generated unseen-class images align closely with seen-class data, confirming the effectiveness of the proposed pipeline.

Table 13: Ablation studies on class quality and data quality. "Acc" denotes the accuracy. "Dis" denotes the distribution distance of the generated unseen-class data and seen-class data.

| Dataset | | Ours | | Class Quality | | | | Data Quality | | | | | |
| | | | | LowSim | | w/o Tree | | Picture | | Photo | | Image | |
| | | Acc ↑ | Dis ↓ | Acc ↑ | Dis ↓ | Acc ↑ | Dis ↓ | Acc ↑ | Dis ↓ | Acc ↑ | Dis ↓ | Acc ↑ | Dis ↓ |
|---|---|---|---|---|---|---|---|---|---|---|---|---|---|
| Caltech | Base | **98.97** | | 98.26 | | 97.93 | | 98.26 | | 98.19 | | 98.06 | |
| | New | **95.85** | **9.99** | 94.32 | 10.49 | 93.67 | 13.16 | 94.21 | 11.84 | 94.11 | 11.95 | 93.78 | 12.12 |
| | H | **97.38** | | 96.25 | | 95.75 | | 96.19 | | 96.11 | | 95.87 | |
| Pets | Base | **96.01** | | 95.85 | | 95.00 | | 95.69 | | 95.43 | | 95.27 | |
| | New | **98.27** | **7.58** | 97.60 | 8.19 | 96.81 | 11.71 | 97.04 | 9.14 | 96.98 | 9.21 | 96.92 | 10.27 |
| | H | **97.12** | | 96.72 | | 95.90 | | 96.36 | | 96.20 | | 96.09 | |
| Cars | Base | **82.93** | | 78.94 | | 77.86 | | 77.99 | | 78.71 | | 78.24 | |
| | New | **80.81** | **8.92** | 75.29 | 9.33 | 73.36 | 13.78 | 74.75 | 10.65 | 75.04 | 10.08 | 74.92 | 10.48 |
| | H | **81.86** | | 77.07 | | 75.54 | | 76.33 | | 76.83 | | 76.54 | |
| Flowers | Base | **98.77** | | 98.29 | | 97.15 | | 97.82 | | 97.91 | | 97.63 | |
| | New | **80.92** | **6.29** | 77.52 | 6.88 | 75.04 | 10.29 | 76.88 | 7.99 | 77.09 | 7.84 | 76.17 | 8.22 |
| | H | **88.96** | | 86.68 | | 84.67 | | 86.09 | | 86.26 | | 85.57 | |
| Food | Base | **91.39** | | 90.90 | | 90.45 | | 90.56 | | 90.60 | | 90.81 | |
| | New | **92.99** | **9.01** | 92.25 | 9.49 | 91.79 | 13.80 | 91.83 | 11.24 | 91.97 | 11.10 | 92.02 | 10.94 |
| | H | **92.18** | | 91.57 | | 91.12 | | 91.19 | | 91.28 | | 91.41 | |
| Aircraft | Base | **48.98** | | 43.88 | | 42.56 | | 42.98 | | 42.80 | | 43.64 | |
| | New | **44.03** | **8.63** | 37.97 | 8.89 | 34.85 | 15.79 | 36.23 | 12.86 | 35.51 | 13.64 | 36.89 | 12.04 |
| | H | **46.37** | | 40.71 | | 38.32 | | 39.32 | | 38.82 | | 39.98 | |
| DTD | Base | **85.53** | | 83.91 | | 81.48 | | 83.22 | | 82.87 | | 83.57 | |
| | New | **71.50** | **7.67** | 65.22 | 8.01 | 57.73 | 8.84 | 63.77 | 8.38 | 62.32 | 8.73 | 63.89 | 8.20 |
| | H | **77.89** | | 73.39 | | 67.58 | | 72.21 | | 71.14 | | 72.41 | |
| EuroSAT | Base | **97.17** | | 94.71 | | 91.05 | | 91.83 | | 92.12 | | 91.62 | |
| | New | **87.90** | **11.48** | 80.49 | 11.71 | 65.28 | 12.39 | 68.08 | 11.88 | 71.67 | 11.82 | 67.41 | 12.07 |
| | H | **92.30** | | 87.02 | | 76.04 | | 78.19 | | 80.62 | | 77.67 | |
| UCF | Base | **89.14** | | 86.97 | | 85.88 | | 86.25 | | 86.66 | | 86.14 | |
| | New | **82.53** | **11.70** | 77.99 | 12.29 | 76.53 | 13.39 | 77.45 | 12.70 | 77.88 | 12.67 | 77.29 | 13.06 |
| | H | **85.71** | | 82.23 | | 80.94 | | 81.61 | | 82.04 | | 81.47 | |

## Seen-Class Data

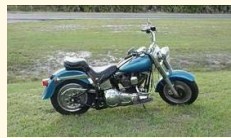 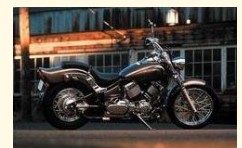 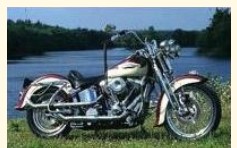

## Generated Seen-Class Data

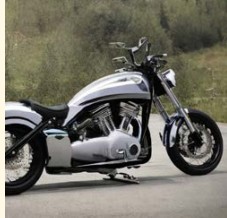 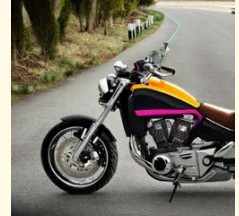 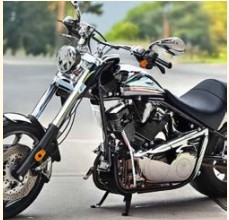

**Generated Seen-Class Data from Our Pipeline**

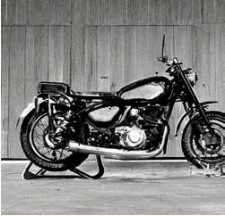 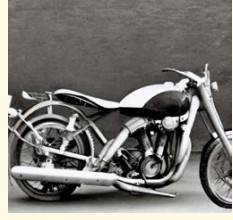 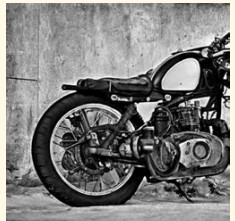

Template 'picture'  Template 'photo'  Template 'image'

**Generated Seen-Class Data from Three Templates**

## Generated Unseen-Class Data

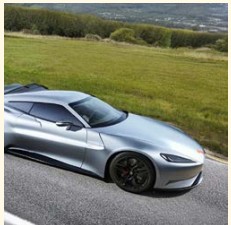 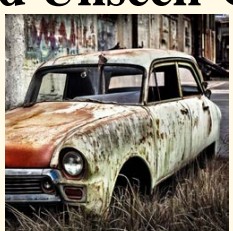 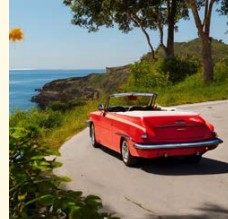

**Generated Unseen-Class Data from Our Pipeline**

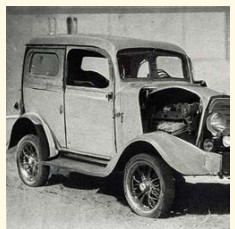 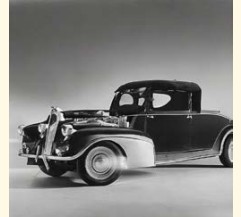 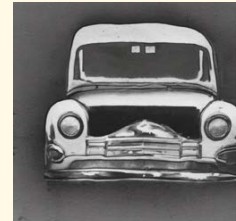

Template 'picture'  Template 'photo'  Template 'image'

**Generated Unseen-Class Data from Three Templates**

Figure 4: Comparison between the images generated with class-domain-wise data generation pipeline and three prompt templates mentioned in ablation studies. The seen class is 'motorbike' and the inferred unseen class is 'car'.

Figure 5: Comparison between the images generated with class-domain-wise data generation pipeline and three prompt templates mentioned in ablation studies. The seen class is 'barrel' and the inferred unseen class is 'drum'.

