# OpenReview forum: "Beyond the Seen: Bounded Distribution Estimation for Open-Vocabulary Learning"
_NeurIPS.cc/2025/Conference — NeurIPS 2025 poster_

### Official Review · Reviewer_gXSZ · 2025-06-22

**Clarity:** 3
**Significance:** 2
**Originality:** 2
**Rating:** 4
**Confidence:** 4

**Summary:**

The paper tackles the open-vocabulary learning challenge by generating synthetic data for unseen classes to address the distribution estimation problem. Theoretical analysis shows that this approach can bound the estimation error (Theorems 1 and 2). The method includes a data generation pipeline using semantic trees and domain info to guide image synthesis, and a distribution alignment algorithm with KL-divergence and MMD losses.  Experiments on 11 datasets demonstrate SOTA performance. Ablations highlight the importance of each component.

**Questions:**

1)	Data Generation Quality: How does the quality of generated unseen-class data impact the overall performance of the method? Could the authors provide more details on the quality metrics used for evaluating the generated data?
2)	Scalability to Large-Scale Unseen Classes: The class predictor uses top-K_0 candidates from a hierarchical tree. How does performance degrade as K_0 increases (e.g., 100+ classes)? Provide metrics (e.g., distribution distance, accuracy) for larger K_0 in the rebuttal.
3)	How robust is the distribution alignment algorithm to noise in the generated data? Could the authors discuss potential improvements to enhance the robustness of the method?

**Ethical Concerns:**

["NO or VERY MINOR ethics concerns only"]

**Final Justification:**

The authors have addressed my previous concerns. I raised my score.

**Limitations:**

Limitations are discussed (Section 8): The Euclidean feature space may not capture complex data geometries. Future work will explore hyperbolic spaces. However:
- Computational Costs: Add inference time/memory metrics (e.g., vs. PromptSRC).
- No discussion of biases in generated data (e.g., via WordNet hierarchies).
Further discussion is needed.

**Quality:**

2

**Strengths And Weaknesses:**

Strengths:
1)	The paper introduces a new approach to open-vocabulary learning by generating unseen-class data, which is a unique combination of data generation and distribution estimation.
2)	The theoretical analysis provides a solid foundation for the proposed method, and the experimental results demonstrate significant improvements over existing methods.
3)	The method addresses a critical challenge in open-vocabulary learning, offering substantial improvements in generalization to unseen classes, which is highly impactful for the research community.
4)	The paper is well-written, with clear problem formulation (Section 3), method description (Section 5), and experimental analysis (Section 6). Theorems are intuitively interpreted.

Weaknesses:
1)	Data Generation Limitations: The quality of generated unseen-class data may be limited by the text-to-image model and the domain information extracted, potentially affecting the overall performance.
2)	Generalization Scope: Experiments focus on image recognition; validation on more complex tasks (e.g., object detection) is needed.
3)	Computational Cost: The pipeline (LLM queries, Stable Diffusion) incurs high inference overhead. Efficiency analysis (e.g., latency vs. performance) is omitted.
4)	Theoretical Assumptions: Theorem 2 assumes p(Y_e)>0, which may not hold for highly dissimilar unseen classes. Sensitivity to this assumption is untested.

---

> ### Author Rebuttal · Authors · 2025-07-31
>
> Thanks for your insightful and thorough review. We will address your concerns one by one.
>
> >**W1:** Data Generation Limitations: The quality of generated unseen-class data may be limited by the text-to-image model and the domain information extracted, potentially affecting the overall performance.
> **Q1:** Data Generation Quality: How does the quality of generated unseen-class data impact the overall performance of the method? Could the authors provide more details on the quality metrics used for evaluating the generated data?
>
> **R:** The quality of generated unseen-class data affects overall performance, as validated in Section 6.4.1. The results (Table 3 in the manuscript) show that higher data quality leads to improved overall performance. The quality metric is the KL divergence between feature distributions of generated unseen-class data and seen-class training data, extracted by a non-tuned CLIP. A smaller divergence indicates higher quality. As shown in Table 3, our method achieves the best generation quality and performance compared to using plain prompts in text-to-image models.
>
> The effectiveness of our method stems from the proposed caption-based domain information generator, which generates high-confidence and diverse domain prompts. With such carefully constructed prompts, current text-to-image models can generate high-quality images [1,2]. To further evaluate the generated image quality, we computed CLIPScore [3], as shown in Table 1. Results show that these images exhibit high semantic quality (CLIPScore >0.35 is considered high quality [4,5,6]). We further conducted a user study and a GPT-4-based evaluation, where both human annotators and GPT-4 independently rated 200 randomly selected images on a 1–5 scale (5 is the highest quality). The resulting average scores of 4.67 (human) and 4.59 (GPT-4) confirm the high quality of the generated images.
>
> **Table 1: CLIPScore of generated images**
> | UCF101  | DTD | SUN397 | Caltech101 | OxfordPets | StanfordCars
> |:-:| :-: | :-: | :-: | :-: | :-: |
> |0.43|   0.42    | 0.43       | 0.43 | 0.44 | 0.42 |
>
> [1] ImagineFSL: Self-Supervised Pretraining Matters on Imagined Base Set for VLM-based Few-shot Learning. CVPR2025.
>
> [2] IS SYNTHETIC DATA FROM GENERATIVE MODELS READY FOR IMAGE RECOGNITION? ICLR2023.
>
> [3] CLIPScore: A Reference-free Evaluation Metric for Image Captioning. EMNLP2021.
>
> [4] Scaling Down Text Encoders of Text-to-Image Diffusion Models. CVPR2025.
>
> [5]  Hrs-bench: Holistic, reliable and scalable benchmark for text-to-image models. ICCV2023.
>
> [6] Photorealistic text-to-image diffusion models with deep language understanding. NeurIPS2022.
>
>
> >**W2:** Generalization Scope: Experiments focus on image recognition; validation on more complex tasks (e.g., object detection) is needed.
>
> **R:**  We focus on open-vocabulary distribution estimation, with extensive experiments on 11 recognition datasets across base-to-new and cross-dataset settings, which sufficiently validate the effectiveness of our method. As our method builds on prompt learning, we follow prior works [1–3] and evaluate primarily on image recognition tasks. The proposed method investigates general principles of distribution estimation, and is agnostic to the specific downstream vision task. We appreciate this suggestion and plan to leverage large pretrained models (e.g., SAM or Grounding-DINO) to provide pseudo annotations and adapt our alignment algorithm to object detection and segmentation tasks accordingly.
>
> [1] Learning to prompt for vision-language models. IJCV2022.
>
> [2] Self-regulating prompts: Foundational model adaptation without forgetting. ICCV2023.
>
> [3] Divergence-enhanced knowledge-guided context optimization for visual-language prompt tuning. ICLR2025.
>
> >**W3:** Computational Cost: The pipeline (LLM queries, Stable Diffusion) incurs high inference overhead. Efficiency analysis (e.g., latency vs. performance) is omitted.
> >**L1:** Computational Costs: Add inference time/memory metrics (e.g., vs. PromptSRC).
>
> **R:** Thank you for the suggestion. We provide more details regarding the computational efficiency of our method, which will be added to the revised version.
>
> The training memory and training time of the component of our method on RTX 4090 (using the Eurosat dataset as an example) are as follows:
> - VLMs (LLaVA-1.5-7B) for generating image captions: 15.65 GB GPU memory and 121.8 seconds for 80 images;
> - LLMs (Llama-3.1-8B-Instruct) for extracting domain information: 16.14 GB GPU memory and 11.2 seconds for 5 prompts;
> - Stable Diffusion for generating images: 7.13 GB GPU memory and 480 seconds for 240 images;
> - Training for models: 6.20 GB GPU memory and 126.2 seconds.
>
> Since these steps run sequentially without needing to load all into memory, the overall method requires at most 16.14 GB and 739.2 seconds. The baseline (PromptSRC) requires 6.12 GB and 101.25 seconds. The added time and memory mainly come from data generation.
>
> At inference time, no extra parameters or computation are introduced. Thus, our runtime (3.8 seconds) and memory (1.84 GB) are identical to the baseline. In summary, although our method requires additional computation time and memory usage, the performance improvement of up to 14% makes this cost worthwhile.
>
> >**W4:** Theoretical Assumptions: Theorem 2 assumes p(Y_e)>0, which may not hold for highly dissimilar unseen classes. Sensitivity to this assumption is untested.
>
> **R:** In practice, $P(Y_{e}) > 0$ always holds without any constraint on $Y_e$, as analyzed below. We first clarify the definitions and relationships of label sets in open environments. In the open-vocabulary setting, by definition, the open-environment label set $Y_{o}$ is the union of seen-class label set $Y_{s}$ and **unseen-class label set $Y_{u}$**, which is mathematically represented as
> $$
> Y_{o} = Y_{s} \cup Y_{u}, Y_{s} \cap Y_{u} =  \emptyset.
> $$This implies that any class in open environments outside the seen class set $Y_{s}$ is considered an unseen class.  Our **predicted unseen-class label set $Y_{e}$** satisfies  $Y_{e} \cap Y_{s} = \emptyset$ and thus $Y_{e} \subset Y_{u}$. The **test  unseen-class label set $Y_{t}$** also satisfies that $Y_{t} \cap Y_{s} = \emptyset$ and thus $Y_{t} \subset Y_{u}$. We clarify that in Theorem 2, **$P(Y_{e})$ is defined as the probability assigned to the set $Y_{e}$ under the distribution over the set $Y_{u}$.** Since $Y_{e} \subset Y_{u}$, it naturally follows that $P(Y_{e}) > 0$. Moreover, even if $Y_{e} \cap Y_{t} = \emptyset$, indicating that $Y_{e}$ may be highly dissimilar to $Y_{t}$,  it still holds that $P(Y_{e}) > 0$. As a result, the assumption is not sensitive, even for highly dissimilar unseen classes. Thank you for pointing this out. We will add a related clarification to the revised version.
>
> >**Q2:** Scalability to Large-Scale Unseen Classes: The class predictor uses top-K_0 candidates from a hierarchical tree. How does performance degrade as K_0 increases (e.g., 100+ classes)? Provide metrics (e.g., distribution distance, accuracy) for larger K_0 in the rebuttal.
>
> **R:** We set $K_0 = 1$ in our main experiments. We analyze the scalability of our method with larger $K_0$  by setting $K_0 = 5, 10, 20$, as shown in Table 2. Results show that our method remains robust across different $K_0$. This robustness stems from our sparse loss computation strategy, which aligns the distribution of seen-class batches with the semantically similar unseen-class distribution, reducing misalignment and exhibiting robustness as $K_0$ increases.
>
>  **Table 2: $K_0$ analysis**
> |    | $K_0 = 1$ | $K_0 = 5$ | $K_0 = 10$ | $K_0 = 20$ |
> |:-:| :-: | :-: | :-: | :-: |
> |Base|   97.17 | 96.64    | 96.43 | 96.24 |
> |New | 87.90  |  87.07    | 85.85 | 85.84|
> |H    | 92.30  | 91.61   | 90.83 | 90.57 |
>
> >**Q3:** How robust is the distribution alignment algorithm to noise in the generated data? Could the authors discuss potential improvements to enhance the robustness of the method?
>
> **R:** Our distribution alignment algorithm is robust to noise in the generated data due to its sparse loss computation strategy. Instead of aligning seen classes with all generated batches, it selectively aligns seen-class distributions with the semantically similar generated unseen-class batch, avoiding noisy alignments and even filtering out low-quality samples, thereby improving robustness. To verify this robustness, we conduct an experiment using incorrect superclasses. Results in Table 3 show that the performance only slightly degraded. This demonstrates the robustness of the algorithm to the noise in generated images.
>
> **Table 3: Robustness evaluation to the noisy semantic tree**
> |   | Correct Superclasses | Wrong Superclasses | PromptSRC (Baseline) |
> |:-:| :-: | :-: | :-: |
> |Base|   97.17    | 96.17       | 92.90 |
> |  New | 87.90    | 86.49       | 73.90 |
> |  H | 92.30      | 91.07       | 82.32 |
>
> A potential direction is to incorporate confidence-aware weighting, down-weighting generated samples with low confidence or high uncertainty during alignment. Another is to iteratively refine the generation and alignment stages, enabling the model to improve data quality based on alignment feedback.
>
> >**L2:** No discussion of biases in generated data (e.g., via WordNet hierarchies). Further discussion is needed.
>
> **R:** Thanks for pointing it out. While biases may exist in WordNet or the training data of pretrained language models, these biases do not undermine our core contributions, as our method is empirically validated to be robust to such imperfections.
>
> To mitigate this issue, we plan to design a multi-expert collaboration strategy that leverages diverse pretrained models and agreement-based selection to reduce dependence and bias on any single model. This discussion will be added to the revised manuscript.

---

> > ### Comment · Reviewer_gXSZ · 2025-08-03
> > **Official Comment by Reviewer gXSZ**
> >
> > Thanks for the detailed responses to my concerns. I will raise my final score.

---

> > > ### Author Response · Authors · 2025-08-03
> > >
> > > Thank you for your thoughtful review and for raising the score! We’re thrilled to know that our response addressed your concerns effectively. Your constructive comments have been essential in improving this work, and we truly appreciate the suggestions you’ve provided.

---

### Official Review · Reviewer_2wK7 · 2025-06-26

**Clarity:** 3
**Significance:** 4
**Originality:** 4
**Rating:** 5
**Confidence:** 4

**Summary:**

This work proposes a method for estimating the distribution of unseen classes in the field of open-vocabulary learning, by generating of samples from unseen classes. It also provides a theoretical analysis of the estimation error bounds. Based on this theory, the paper propose an open-vocabulary learning method that constructs unseen-class samples to maximize the posterior probability, achieving significant performance improvements.

**Questions:**

(1) Include the supplementary materials and adjust the content between the main text and the supplementary materials accordingly.

(2) In Section 5.1 "Class-Domain-Wise Data Generation Pipeline," the data generation process control could be made more rigorous.

(3) The sparse loss computation strategy is well-designed, but it could be explored in greater depth.

**Ethical Concerns:**

["NO or VERY MINOR ethics concerns only"]

**Final Justification:**

I maintain my previous rating.

I have read the author rebuttal and found that it addressed my requests. I am also satisfied with the authors’ responses in the discussions with other reviewers. Considering these points, I have no remaining concerns and therefore keep my current rating.

**Limitations:**

yes

**Quality:**

4

**Strengths And Weaknesses:**

Strengths:

The theoretical analysis is solid, the method is cleverly designed, and the results are remarkable. The experiments are thorough, and the paper is logically rigorous.

Weaknesses:

(1) The supplementary materials need to be included, as too many important details have been omitted.

(2) The paper lacks a framework diagram and sample visualizations, making it difficult for readers to understand.

---

> ### Author Rebuttal · Authors · 2025-07-31
>
> Thanks for your insightful and thorough review. We will address your concerns one by one.
>
> >**W1:** The supplementary materials need to be included, as too many important details have been omitted. The paper lacks a framework diagram and sample visualizations, making it difficult for readers to understand.
> >**Q1:** Include the supplementary materials and adjust the content between the main text and the supplementary materials accordingly.
>
> **R:** Thanks for your suggestions. We will add the necessary important details in the supplementary materials to the revised version, including the framework diagram and sample visualizations.
>
> Specifically, we will add the formulation of the class-domain-wise data generation pipeline (shown in Figure 1 of the supplementary materials) to Section 5. The sample introductions and specific details of the proposed method presented in Sections 4.1 and 4.2 of the supplementary materials will be merged into Section 5.1 of the revised version. Moreover, we will add the visualization of extracted domain information and generated unseen-class data (shown in Section 8 in the supplementary materials) to Section 6 of the revised version.
>
> To accommodate these changes within the page limit, we will reduce the content related to the ablation studies (Section 6.4). Some detailed descriptions and results will be moved to the supplementary materials accordingly.
>
> >**Q2:** In Section 5.1 "Class-Domain-Wise Data Generation Pipeline," the data generation process control could be made more rigorous.
>
> **R:** The rigorous control over the data generation process is indeed essential to ensure the quality and alignment of the synthetic unseen-class data, further leading to accurate distribution estimation.
>
> To this end, our current pipeline already incorporates multiple design elements aimed at minimizing noise and improving distribution alignment. The unseen class predictor utilizes a cosine similarity-based candidate selection strategy to effectively filter irrelevant or overly distant candidate unseen classes. The caption-based domain information generator extracts style and scene attributes from seen-class data to guide image generation more accurately. Moreover, we use multiple class-specific captions to extract diverse domain information, thereby enhancing the diversity of generated images and reducing unnecessary semantic noise.
>
> Thanks for your suggestions. In future work, we plan to enhance the pipeline by constructing confidence thresholds and entailment constraints on semantic tree construction for robust unseen class predictions, and introducing CLIP-based filtering to discard generated images with low semantic consistency.
>
> >**Q3:** The sparse loss computation strategy is well-designed, but it could be explored in greater depth.
>
> **R:** Thank you for the positive feedback on the proposed sparse loss computation strategy. We design it to alleviate distribution misalignment caused by per-batch variability. To further explore and validate its effectiveness, we conduct additional experiments, including ablations and hyperparameter analysis. Specifically, we compared our method, which aligns the distributions of stored seen-class batches with those of the top-$K_3$ most similar unseen-class batches, with two alternative strategies:
> 1. Random batch matching: seen-class distributions are aligned with randomly sampled unseen-class batches;
> 2. Least-similar batch matching: alignment is performed with the top-$K_3$ unseen-class batches that are least similar (i.e., worst-case alignment).
>
> The results shown in Table 1(a) clearly demonstrate that our method significantly outperforms both alternatives, confirming the effectiveness of the proposed sparse loss computation strategy.
>
>
> **Table 1(a): Alignment strategy analysis**
> |   | Our method | Random batch matching | Least-similar batch matching |
> |:-:| :----: | :--------: | :--------: |
> |Base|   97.17 |  96.43   | 96.02 |
> |New | 87.90  |  84.69   | 82.87 |
> |H    | 92.30  | 90.18    | 88.96 |
>
>
> Additionally, we conduct a hyperparameter study on the value of $K_3$ (the number of most similar unseen-class distributions used for alignment). Experimental results presented in Table 1(b) show that performance remains robust as $K_3$ varies.
>
> **Table 1(b): K_3 Analysis**
> |   | $K_3 = 1$ | $K_3 = 2$ | $K_3 = 3$ |
> |:-:| :----: | :--------: | :--------: |
> |Base|   97.17 | 96.33    | 96.60 |
> |New | 87.90  | 85.36   | 86.31 |
> |H    | 92.30  | 90.51    | 91.16 |
>
>
> In future work, we aim to explore more advanced variants of this strategy, including adaptive memory weighting (e.g., attention mechanisms) and class-aware matching. These directions could further improve the robustness and generalization of distribution alignment in open environments.

---

### Official Review · Reviewer_YUQg · 2025-07-02

**Clarity:** 3
**Significance:** 2
**Originality:** 3
**Rating:** 5
**Confidence:** 4

**Summary:**

This work proposes a novel approach to the problem setting of open-vocabulary learning. The key idea is to generate unseen-class data for distribution estimation in open environments. The work provides the theoretical bounds that indicate that the gap between the joint distributions of generated unseen-class data and unseen-class data can be narrowed down by reducing the distance between the joint distributions of generated unseen-class data and seen-class data. Empirically, this work demonstrates state-of-the-art performance on 11 datasets, outperforming prompt-based baselines, especially in base-to-new and cross-dataset generalization.

**Questions:**

1. The main concern is according to Weakness 1. The assumption (a) is commonly used in the area of open-vocabulary learning, so it is fair that this paper also uses this assumption. Under this assumption, it is important to show that there is a potential to extend the method to the truly unknown classes. However, in the paper, even if the authors claim that they will extend our analysis and method to hyperbolic spaces to capture the complex geometric properties, it is hard to see why it is possible to extend the current method. For example, for the truly unknown classes, how can the LLMs and WordNet predict them? Moreover, even if a new word is predicted, how can the text-to-image model generate the synthetic data, or can we still trust the synthetic data?
2. Another concern is according to Weakness 1, but assumption (b). In general, the basic assumption of open-vocabulary learning is that the model has never seen the visual examples of the unseen classes during training. However, the pre-trained text-to-image models likely saw examples of unseen classes during training. Is this a kind of data leakage risk?

Personally, I appreciate that this work shows the theoretical bounds of the method. However, the main concerns make me worry about the feasibility of applying this method in a real open-world environment. If the author addresses my concern, I will improve the score to 5.

**Ethical Concerns:**

["NO or VERY MINOR ethics concerns only"]

**Final Justification:**

My main concern is the data leakage problem. The authors address my main concern by explaining the problem setting using the previous works and showing the results of the EuroSAT dataset, where the class overlap is 0%.

**Limitations:**

Yes.

**Quality:**

3

**Strengths And Weaknesses:**

Strengths:
1. This work shows the theoretical bounds of estimation errors in joint and posterior distributions when using generated unseen-class data, which provides theoretical support to the method.
2. The method demonstrates strong empirical results across 11 diverse datasets. In addition, this work also conducted sufficient ablation studies.

Weaknesses:
1. The proposed method is based on two very strong assumptions: (a) The unseen classes are semantically close to the seen ones so that the WordNet or LLMs can predict the name of the unseen classes. (b) The images of the predicted unseen classes can be generated by the pretrained text-to-image model.
2. There is no discussion of computational efficiency in the paper. Since the method depends on other models like some LLMs and Stable Diffusion, the computational cost can also be much higher.

---

> ### Author Rebuttal · Authors · 2025-07-31
>
> Thanks for your insightful and thorough review. We will address your concerns one by one.
>
> >**W1:** There is no discussion of computational efficiency in the paper. Since the method depends on other models like some LLMs and Stable Diffusion, the computational cost can also be much higher.
>
> **R:** Thank you for the suggestion. We provide more details regarding the computational efficiency of our method, which will be added to the revised version.
>
> The training memory and training time of the component of our method on RTX 4090 (using the Eurosat dataset as an example) are as follows:
> - VLMs (LLaVA-1.5-7B) for generating image captions: 15.65 GB GPU memory and 121.8 seconds for 80 images;
> - LLMs (Llama-3.1-8B-Instruct) for extracting domain information: 16.14 GB GPU memory and 11.2 seconds for 5 prompts;
> - Stable Diffusion for generating images: 7.13 GB GPU memory and 480 seconds for 240 images;
> - Training for models: 6.20 GB GPU memory and 126.2 seconds.
>
> Since these steps run sequentially without needing to load all into memory, the overall method requires at most 16.14 GB and 739.2 seconds. The baseline (PromptSRC) requires 6.12 GB and 101.25 seconds. The added time and memory mainly come from data generation.
>
> At inference time, no extra parameters or computation are introduced. Thus, our runtime (3.8 seconds) and memory (1.84 GB) are identical to the baseline. In summary, although our method requires additional computation time and memory usage, the performance improvement of up to 14% makes this cost worthwhile.
>
> >**Q1:** The main concern is according to Weakness 1. The assumption (a) is commonly used in the area of open-vocabulary learning, so it is fair that this paper also uses this assumption. Under this assumption, it is important to show that there is a potential to extend the method to the truly unknown classes. However, in the paper, even if the authors claim that they will extend our analysis and method to hyperbolic spaces to capture the complex geometric properties, it is hard to see why it is possible to extend the current method. For example, for the truly unknown classes, how can the LLMs and WordNet predict them? Moreover, even if a new word is predicted, how can the text-to-image model generate the synthetic data, or can we still trust the synthetic data?
>
> **R:** Our future work to explore hyperbolic spaces is aimed at addressing the issue of inconsistency between the geometric structure of data and the Euclidean spaces used in models, and leveraging the hyperbolic manifolds to obtain the more robust data representation. This future work is not intended to break assumption (a) (the unseen classes are semantically close to the seen ones so that the WordNet or LLMs can predict the name of the unseen classes) to extend to truly unknown classes.
>
> In fact, compared to existing methods, our method exhibits greater potential to handle truly unknown classes. From the conducted experiments, we observe that
> 1. In cross-dataset experiments, seen classes from ImageNet are semantically far from target classes such as those in EuroSAT or DTD. Our method still achieves 9.48% and 1.4% improvements, respectively.
> 2. In base-to-new experiments, on EuroSAT, we surprisingly find that our predicted unseen classes are entirely different from the test unseen classes, with no overlap between them. Our method still achieves a 14% improvement over the baseline.
>
> These improvements stem from that our method can provide accurate and effective distribution estimation in open environments, without a strong constraint that the unseen classes are semantically close to seen classes.
>
> In future work, to extend our method to support truly unknown classes, such as newly cartoon characters and emerging vehicle models, we plan to integrate Retrieval-Augmented Generation (RAG) mechanisms that dynamically retrieve emerging classes from external sources, enabling the prediction pipeline to go beyond static knowledge in WordNet and LLM. Then, we plan to retrieve relevant images from the Internet as in-context examples to enhance the ability of the model to generate up-to-date images, enabling the model to adapt in real time to new concepts.
>
> >**Q2**: Another concern is according to Weakness 1, but assumption (b). In general, the basic assumption of open-vocabulary learning is that the model has never seen the visual examples of the unseen classes during training. However, the pre-trained text-to-image models likely saw examples of unseen classes during training. Is this a kind of data leakage risk?
>
> **R:** We consider that there is no risk of data leakage in our method. The key reason is that, based on related literature, the core constraint in open-vocabulary recognition is that the model should not have access to labels or explicit clues for unseen classes during training [1, 2], rather than "the model has never seen the visual examples of the unseen classes during training". For example, the work [3] explicitly explores ways to utilize pretrained VLMs to obtain image-level clues for unseen classes; some methods [4,5,6] also leverage large-scale additional image data to tune CLIP for improvement in open-vocabulary tasks, which inevitably introduces overlap with categories in the test set. Especially, the work [4] conducts prompt learning on real images of 21,841 classes from ImageNet21K, and we find that the class overlap rate between the training set and the test set reaches up to 23%. These observations collectively suggest that the core constraint in open-vocabulary learning lies in the absence of category-level clues about unseen classes during training, rather than a strict prohibition of any form of exposure to their visual instances.
>
>
> The difference between our method and the above methods is that we utilize synthetic images whose categories are predicted by LLMs or WordNet, whereas prior work relies on real images collected manually. In our method, while predicted classes may match the test classes, we find that the class overlap rate across datasets is at most 6% and at least 0%, being significantly lower than that in previous work (23%) [4]. Apart from this, our method indeed does not use any images or information from the test set. Therefore, we believe that there is no risk of data leakage in our method.
>
>
> Moreover, we added an additional analysis in our experiments. On the EuroSAT dataset, where the class overlap is 0% (i.e., no predicted unseen classes match the test classes), our method still achieves up to a 14% improvement. Additionally, we conduct an experiment by removing all overlapping test classes. Evaluation on the remaining test classes shows that our method continues to perform strongly, achieving up to an 8.29% improvement on the DTD dataset. These confirm that the improvements of our method are not attributed to the class overlap, which the reviewers refer to as "data leakage".
>
>
> [Reference]
>
> [1] Ma Z, Zhang S, Wei L, et al. Ovmr: Open-vocabulary recognition with multi-modal references[C]//Proceedings of the IEEE/CVF Conference on Computer Vision and Pattern Recognition. 2024: 16571-16581.
>
> [2] He S, Guo T, Dai T, et al. Open-vocabulary multi-label classification via multi-modal knowledge transfer[C]//Proceedings of the AAAI conference on artificial intelligence. 2023, 37(1): 808-816.
>
> [3] Zang Y, Li W, Zhou K, et al. Open-vocabulary detr with conditional matching[C]//European conference on computer vision. Cham: Springer Nature Switzerland, 2022: 106-122.
>
> [4] Ren S, Zhang A, Zhu Y, et al. Prompt pre-training with twenty-thousand classes for open-vocabulary visual recognition[J]. Advances in Neural Information Processing Systems, 2023, 36: 12569-12588.
>
> [5] Xu Y, Zhang M, Fu C, et al. Multi-modal queried object detection in the wild[J]. Advances in Neural Information Processing Systems, 2023, 36: 4452-4469.
>
> [6] Kaul P, Xie W, Zisserman A. Multi-modal classifiers for open-vocabulary object detection[C]//International Conference on Machine Learning. PMLR, 2023: 15946-15969.

---

> > ### Comment · Reviewer_YUQg · 2025-08-01
> >
> > I appreciate the authors' detailed rebuttal, which has addressed my main concerns, and hence I will update my score to a positive rating.

---

> > > ### Author Response · Authors · 2025-08-01
> > >
> > > Thank you for your thoughtful review and for raising the score! We’re thrilled to know that our response addressed your concerns effectively. Your constructive comments have been essential in improving this work, and we truly appreciate the suggestions you’ve provided.

---

### Official Review · Reviewer_BCAC · 2025-07-03

**Clarity:** 3
**Significance:** 3
**Originality:** 3
**Rating:** 5
**Confidence:** 3

**Summary:**

This paper introduces a novel framework for open-vocabulary learning by addressing the challenge of distribution estimation in open environments. Specifically, the method consists of two main components: (1) a class-domain-wise data generation pipeline that generates unseen-class data under the guidance of a hierarchical semantic tree and domain information inferred from the seen-class data, and (2) a distribution alignment algorithm that estimates and maximizes the posterior probability to enhance generalization in open-vocabulary learning. Experiments on benchmark datasets show significant improvements over existing methods.

**Questions:**

1. Alignment is performed on the top K3 most similar distributions. Would such a setting ignore the long-tail distribution categories?
2. How to handle the noisy semantic tree and noisy generated noisy images?

**Ethical Concerns:**

["NO or VERY MINOR ethics concerns only"]

**Final Justification:**

The authors have addressed my previous concerns. I keep my score.

**Limitations:**

Yes

**Quality:**

3

**Strengths And Weaknesses:**

Strengths:
1. The paper is well-structured, logically organized and easy to follow. It shows comprehensive analyses of the motivation of the studied problem and the proposed method.

2. The paper provides theoretical results and proofs, which enhances the depth of understanding of the algorithm's performance.

3. Extensive experiments on various datasets are conducted to prove the effectiveness of the proposed approach. The empirical results demonstrate superior or comparable performance to the mentioned methods.

Weaknesses:

1.The corresponding hyperparameters are fixed across all datasets and benchmarks. It is suggested that the authors briefly discuss parameter sensitivity.

---

> ### Author Rebuttal · Authors · 2025-07-31
>
> Thanks for your insightful and thorough review. We will address your concerns one by one.
>
> > **W1:** The corresponding hyperparameters are fixed across all datasets and benchmarks. It is suggested that the authors briefly discuss parameter sensitivity.
>
> **R:** Thanks for your suggestions. We add experiments to analyze the sensitivity of hyperparameters $K_0, K_1, K_2, K_3$ in the data generation pipeline and $\alpha, \beta$ in the loss function. Specifically, we conduct experiments on the EuroSAT dataset by adjusting these hyperparameters. Results presented in Table 1 show that the performance remains relatively stable with varying hyperparameter values, indicating that the method is only minimally sensitive to hyperparameter variation.
>
>
>  **Table 1(a): $K_0$ analysis**
> |   | $K_0 = 1$ | $K_0 = 5$ | $K_0 = 10$ | $K_0 = 20$ |
> |:-:| :----: | :--------: | :--------: | :--------: |
> |Base|   97.17 | 96.64    | 96.43 | 96.24 |
> |New | 87.90  |  87.07    | 85.85 | 85.84|
> |H    | 92.30  | 91.61   | 90.83 | 90.57 |
>
>
> **Table 1(b): $K_1$ analysis**
> |   | $K_1 = 8$ | $K_1 = 6$ | $K_1 = 4$ |  $K_1 = 2$ |
> |:-:| :----: | :--------: | :--------: | :--------: |
> |Base|   97.17 | 97.02    | 96.95 | 96.83 |
> |New | 87.90  |  86.31    | 86.0 | 85.85 |
> |H    | 92.30  | 91.35    | 91.15 | 91.01 |
>
>
>
>
> **Table 1\(c\): $K_2$ analysis**
> |   | $K_2 = 3$ | $K_2 = 2$ | $K_2 = 1$ |
> |:-:| :----: | :--------: | :--------: |
> |Base|   97.17 | 96.92    | 96.79 |
> |New | 87.90  |   86.59    | 85.31 |
> |H    | 92.30  | 91.47   | 90.69 |
>
>
>
> **Table 1(d): $K_3$ analysis**
> |   | $K_3 = 1$ | $K_3 = 2$ | $K_3 = 3$ |
> |:-:| :----: | :--------: | :--------: |
> |Base|   97.17 | 96.33    | 96.60 |
> |New | 87.90  | 85.36   | 86.31 |
> |H    | 92.30  | 90.51    | 91.16 |
>
> **Table 1(e): $\alpha$/$\beta$ analysis $(\alpha + \beta = 1)$**
> | α | β | Base | New | H |
> |----|----|------|------|------|
> | 0.3 | 0.7 | 96.88 | 86.15 | 91.20 |
> | 0.4 | 0.6 | 96.86 | 86.26 | 91.25 |
> | 0.5 | 0.5 | 97.17 | 87.90 | 92.30 |
> | 0.6 | 0.4 | 96.81 | 87.06 | 91.68 |
>
>
> > **Q1:** Alignment is performed on the top K3 most similar distributions. Would such a setting ignore the long-tail distribution categories?
>
> **R:** Following the baseline and most prompt learning methods, our experiments focus on settings with balanced data distributions, where the long-tail issue does not arise. Under these conditions, performing alignment on the top $K_3$ most similar distributions is reasonable and effective.
>
> Thanks for your suggestions. To further investigate, we construct a long-tail distribution setting by removing a portion of samples from selected classes and conduct comparative experiments (with the number of samples per class being 16, 16, 16, 10, and 2, respectively). In this setting, we evaluate the performance of our current top $K_3$ alignment strategy against a sample duplication strategy designed for long-tail settings. The results in Table 2 demonstrate that in the long-tailed setting, the sample duplication strategy can improve performance.
>
> **Table 2: Experiments on long-tailed benchmark**
>   |   | Our method (with top $K_3$ alignment) | Our method (with top $K_3$ alignment + sample duplication) |
>   |:-:| :-------------------------------: | :------------------------------------------------------: |
>   |Base| 95.79 | 97.0 |
>   |New | 85.39 | 87.39 |
>   |H    | 90.29 | 91.94 |
>
> >**Q2:** How to handle the noisy semantic tree and noisy generated noisy images?
>
> **R:** We separately analyze the robustness of our method to noisy semantic trees and noisy generated images. Below are details.
>
>
> - Noisy semantic trees. The hierarchy-guided unseen class predictor can mitigate the effect of noisy semantic trees through a cosine similarity-based candidate selection strategy, which effectively filters out irrelevant or overly distant candidate unseen classes.
>
> 	To verify the robustness, we conduct an experiment where we select incorrect superclasses on the EuroSAT dataset. Results in Table 3 show that the performance slightly degrades, demonstrating the robustness of our method.
>
> 	**Table 3: Robustness evaluation to the noisy semantic tree**
> 	|   | Correct Superclasses | Wrong Superclasses | PromptSRC (Baseline) |
> 	|:-:| :-----------: | :-----------: | :-----------: |
> 	|Base|   97.17    | 96.17       | 92.90 |
> 	|  New | 87.90    | 86.49       | 73.90 |
> 	|  H | 92.30      | 91.07       | 82.32 |
>
> - Noisy generated images. The caption-based domain information generator can mitigate the effect of noisy generated images by extracting diverse and class-relevant domain attributes (e.g., styles, scenes) from seen-class data to guide image generation more accurately.
>
> 	To evaluate the quality of the generated images, we compute CLIPScore on six datasets, as shown in Table 4. All results exceed the threshold of 0.35, which indicates high semantic quality [1,2]. We further conduct a user and GPT-based evaluation by rating 200 randomly sampled images on a 1–5 scale, where 5 denotes the highest quality. The average scores are 4.67 from human annotators and 4.59 from GPT-4. These results confirm the high visual and semantic quality of the generated images.
>
> 	**Table 4: CLIPScore of generated images**
> 	| UCF101  | DTD | SUN397 | Caltech101 | OxfordPets | StanfordCars
> 	|:-:| :-----------: | :-----------: | :-----------: | :-----------: | :-----------: |
> 	|0.43|   0.42    | 0.43       | 0.43 | 0.44 | 0.42 |
>
>
> - Furthermore, the proposed sparse loss computation strategy (Sec. 5.2) also contributes to handling both types of noise by selectively aligning the most relevant distributions, thereby mitigating the effect of noise during training.
>
> [Reference]
>
> [1] Wang L, Liu D, Liu X, et al. Scaling Down Text Encoders of Text-to-Image Diffusion Models[C]//Proceedings of the Computer Vision and Pattern Recognition Conference. 2025: 18424-18433.
>
> [2] Bakr E M, Sun P, Shen X, et al. Hrs-bench: Holistic, reliable and scalable benchmark for text-to-image models[C]//Proceedings of the IEEE/CVF International Conference on Computer Vision. 2023: 20041-20053.

---

### Note · Authors · 2025-08-13

Dear Reviewers (BCAC, YUQg, 2wK7, gXSZ) and Area Chair:

We sincerely thank you for your detailed reviews, valuable feedback, and thoughtful discussions throughout the review process. All reviewers have given our paper positive scores, and we are truly grateful for your recognition. We especially appreciate the AC’s support in facilitating productive discussions.

**Summary of Recognized Strengths:** We are grateful for the reviewers’ recognition of our work and its key contributions:
- Well-structured paper with clear motivation and method description (BCAC, 2wK7, gXSZ)
- Solid theoretical analysis with proofs (all reviewers)
- Novel open-vocabulary method via unseen-class data generation (gXSZ)
- Strong empirical analysis on diverse datasets with thorough ablations (all reviewers)
- Significant improvements and better generalization to unseen classes (all reviewers)

**Summary of Rebuttal Efforts:** Through our responses, we have fully addressed all substantive concerns raised by the reviewers. Specifically, we
- Demonstrate that the method is not sensitive to hyperparameter variations
- Analyze the effectiveness in a long-tailed setting
- Analyze the robustness of our method to the noisy generated data
- Clarify the quality of generated data using CLIPScore, user study, and GPT evaluation
- Conduct efficiency comparisons across methods
- Analyze the potential of our method for handling "truly unknown classes"
- Eliminate the reviewer's concern about the risk of data leakage
- Explain that the assumption in theorems is not sensitive

**Summary of Planned Revisions for Camera-Ready:** We will add additional discussions and experiments conducted during rebuttal to the revised version. Specifically, we will
- Add the necessary details to the revised version, including the framework diagram and sample visualizations
- Add discussions on the robustness of our method, the quality of the generated data, the potential of our method for handling “truly unknown” classes, the effectiveness of the sparse loss computation strategy, and the generality of the assumptions
- Add experiments on hyperparameter analysis, evaluation of generated image quality, and efficiency comparison

Thank you again for your valuable contributions to strengthening this work.

Best regards,

Authors

---

### Decision · Program_Chairs · 2025-09-17

**Decision:**

Accept (poster)

**Comment:**

The paper addresses open-vocabulary learning: models are required to generalise beyond seen classes. The paper approximates unseen-class distributions with synthetic samples and computes an upper bound on the "estimation error". The authors propose a class-domain-wise data generation pipeline and a distribution alignment algorithm. Across 11 datasets, the method achieves significant improvements over the state-of-the-art prompt-based baselines.

Strengths. The paper provides theoretical rationale supporting the superiority of the proposed approach (Theorems 1 and 2). Introduces the idea of generating unseen-class data guided by semantic hierarchy and domain information, not just prompts. The idea is compared against suitable baselines on 11 datasets, with both base-to-new and cross-dataset settings. Ample ablations (class quality, data quality, alignment strategies). Addresses noise in semantic hierarchies and generated data, shows insensitivity to hyperparameters, and evaluates long-tailed distributions. Shows consistent improvements over baselines, with especially large margins in challenging datasets (EuroSAT, DTD, Aircraft).

Weaknesses. The work relies on the assumption that unseen classes are semantically close to seen ones and assumes text-to-image models can generate reasonable unseen-class data. The pipeline involves LLMs and diffusion models; efficiency analysis was absent in the original submission. Evaluations are restricted to recognition tasks; object detection and segmentation remain untested.

Rebuttal & discussion. The authors successfully addressed remaining concerns for the reviewers. All reviewers expressed satisfaction post-rebuttal, with improved or maintained positive scores.

Final recommendation: Accept as poster. Key reasons: good problem, novel method, good results.